# Design and Development of a Mathematical Model for an Industrial Process, in a System Dynamics Environment

Jorge Manuel Barrios Sánchez, Roberto Baeza Serrato * and Marco Bianchetti

Departamento de Estudios Multidisciplinarios, División de Ingenierías, Campus Irapuato-Salamanca, Universidad de Guanajuato, Yuriria 38944, Guanajuato, Mexico
* Correspondence: r.baeza@ugto.mx

**Abstract:** This research proposes a methodology based on control engineering, transforming the simulation model of system dynamics into a mathematical model expressed as a system transfer function. The differential equations of a time domain present in the Forrester diagram are transformed into a frequency domain based on the Laplace transform. The conventional control engineering technique is used to present and reduce the dynamic system in a block diagram as a mechanism for determining the structure of the system. The direct path equation and the feedback equation are determined to obtain mathematical models that explain the trajectory of the behavior of each state variable through a transfer function in response to the different inputs of the system. The research proposal is based on presenting an alternative of analytical validation for more robust decision-making to systems dynamics models, based on the explanation of the system structure through a transfer function and its analysis of stability and external controllability for the system dynamics model under study. The results are visually analyzed in a root diagram.

**Keywords:** system dynamic; Forrester; block diagram; differential equations; transfer function; system stability

## 1. Introduction

System Dynamics (SD) is a technique for constructing simulation models, at the service of a fundamental idea; the structure of causal relations between the elements of a system constitutes the main cause of its behavior. The SD further assumes that such a relationship structure forms a network of re-feeding loops that includes non-linear relationships and delays. The main result of a model is to obtain the time trajectories of all its variables for the simulation period that has been defined [1]. SD is based on the use of two types of diagrams, Causal Diagrams and Forrester Diagrams, which have their origin in the General Theory of Systems and are in fact like the two sides of the same coin. The Causal Diagram is in general a preliminary step to the construction of a Forrester Diagram, which serves to simulate the model in the PC, allows us to verify the coherence of our hypotheses, analyze the behavior of the system, and finally simulate different policies, so that the results of the model help to better solve the problem we are analyzing [2].

SD is responsible for studying the feedback of the system data and the means by which the models are used for the establishment and approach of problems, which are carried out with the purpose of showing the structure of the organization and observing how these variables interact, to simulate and model complex problems [3]. Much of the art of SD modeling involves discovering and representing feedback processes, which, along with stock structures, flow, auxiliary variables, time delays and nonlinearities, determine the dynamics of a system. You can imagine that there is an immense range of different feedback processes and other structures to master before you can understand the dynamics of complex systems. In fact, more complex behaviors are usually derived from the interactions (feedback) between system components, not from the complexity of the components

themselves [4]. All dynamics arise from the interaction of only two types of feedback loops, positive (or self-reinforcing) and negative (or self-correcting) loops. Positive loops tend to reinforce or amplify anything that is happening in the system, while swing loops or negatives counteract and oppose change [5]. System Dynamics (SD) is a theory that analyzes dynamic systems (DS) based on simulation models. Control engineering analyzes dynamic systems (DS) based on mathematical models.

The design, development and simulation of SD models in various areas of knowledge is of great importance and relevance in identifying and making explicit the various behaviors based on feedback loops and interactions. This involves making decisions based on simple explanations of the observed results, to improve them, as described by each of the previous authors. However, for a better understanding of the model and better decision-making, it is important to analyses the model of system dynamics with a theoretical and mathematical basis to represent, explain and control the behaviors of the studied variables.

This research proposes a methodology based on control engineering, transforming the simulation model of system dynamics into a mathematical model expressed as a system transfer function. It is important to note that currently the theory of SD does not make use of the necessary mathematical analysis that is used in the theory of control engineering to design control systems. Both theories analyze dynamic systems from their perspective, which are complementary to strengthen the decision-making of SD simulation models, as proposed in this research. In summary, SD analyzes dynamic systems based on a simulation model where multiple variables interact to analyze their behavior, and control engineering analyzes dynamic systems obtaining mathematical models to design control systems.

Simulation models in SD work through differential equations in flow variables. The differential equations of a time domain present in the Forrester diagram are transformed into a frequency domain based on the Laplace transform to facilitate obtaining the output equation, and explain the behavior of state variables in the simulation model. The conventional control engineering technique is used to present and reduce the SD model in a block diagram as a mechanism for determining the structure of the system. The direct path equation and the feedback equation are determined to obtain mathematical models that explain the particular trajectory of the behavior of each state variable through a transfer function in response to the different inputs of the system, in relation to a ratio, which relates the response of a system to an input or excitation signal.

The output signal and the input signal of the dynamic system are represented by algebraic polynomials in the frequency domain. The roots of the denominator polynomial represent the poles of the transfer function to define the stability of the external dynamics of the system through their interpretation in a diagram of roots or poles and zeros. The roots of the numerator polynomial represent the zeros of the transfer function to weight the system's response to the input or excitation signal.

Having the construction and understanding of the SD model, currently the validation of the models is performed by the means of a sensitivity analysis, which consists of a systematic study of how the conclusions of a model are affected by possible variations in parameter values and the functional relationships it includes. The simplest way to perform the analysis is to modify the numerical values of each of its parameters. For this purpose, the value of the parameter whose sensitivity is to be studied is increased by a certain percentage, and the extent to which this variation affects the conclusions of the model (the trajectories it generates) is analyzed. Manual sensitivity analysis requires changing the value of a constant (or several constants at once) and simulating, changing the value of the constant again and simulating again, and repeating this action many times to achieve a spectrum of output values [6].

Today, there are thousands of important applications in the environmental, business, social and scientific sector, developing SD simulation models with a great impact at industrial and academic level, such as [7–22]. The developed models of SD presented in the field of energy, medicine, agriculture, population development, industry 4.0 and sustainability are of great value. In the validation of all the models cited, they use the technique of sensi-

tivity analysis to validate the models, generating multiple scenes and changes in critical variables subjectively, and analyzing changes in their behavior. Since there are multiple variables and interactions in the systems dynamics models, and only the interpretation of the results obtained, it is difficult and complex to identify which parameters should be chosen to perform the sensitivity analysis.

It is evident that an area of opportunity exists to define a mathematical model to identify parameters that affect behavior in a simple, analytical and objective way, rather than the current subjective choice. The present research proposes an analytical alternative of analysis for SD models, which is able to explain the behaviors of state variables, obtaining a mathematical model that explains its external dynamics and controls the particular trajectory of each behavior, instead of just observing the response without understanding whether or not there is any relation to the output in relation to the input of the model under study. In addition, by determining the transfer function, it is possible to analyze and monitor the response to multiple changes in the inputs to the system, the fundamental reason and motivation of this research project.

The validation, analysis and explanation of dynamic systems in control engineering is based on determining the transfer function, in order to analyze the external stability of the system based on the canonical decomposition of controllable and observable characteristic modes at the poles or eigenvalues of the transfer function, and to predict the output behavior at various changes in the determined inputs in a linear combination of the coefficients of the zeros by weighting the exponential response of the poles or eigenvalues of the polynomials of the transfer function, which enter the system and do not depend on any internal system variable. The research proposal is based on presenting an alternative of analytical validation for more robust decision-making to systems dynamics models, based on the explanation of the system structure through a transfer function, and its analysis of stability and external controllability for the system dynamics model under study. The results are visually analyzed in a root diagram.

The link between system structure and dynamic behavior is one of the elements defined in the system dynamics paradigm. Recently, rigorous mathematical and systematic methods have been explored in this link that has been initiated and has become available (Kapman-Oliva). Although several methods have recently been proposed to understand the causes of the model's behavior, formal model analysis remains an important and challenging area in the SD [23].

A SD model can be reduced to a linearized system of time-invariant differential equations and analyzed with the state space technique through a state Equation (1) and an output Equation (2).

$$\dot{X} = Ax + Bu \tag{1}$$

$$y = Cx \tag{2}$$

where Equation (1) explains the internal dynamics and controllability of the system. The internal dynamics of the system is analyzed with the first term of the equation, based on explaining the stability of the model, and the relationship with the system inputs is analyzed with the second term to define the controllability of the system. Equation (2) determines the observability of the system by obtaining response measurements in relation to the internal dynamics of the system.

Recent studies have proposed several analytical methods based on the theory of linear systems and control engineering, particularly the analysis of values and eigenvectors as a pragmatic support. This is intended to be a link to explain the dominant structure of the system based on the dynamic behavior of the loops present in the DS model, such as the studies carried out by [24–33], which investigate the impact of system dominance and elasticity through the determination of vectors and eigenvalues, noting the importance of their influence on the different feedback loops present in the system dynamics model. In these investigations, they reduce the model of system dynamics to a linearized system of first-order time-invariant differential equations, establishing the matrix of partial

derivatives A of order nXn in order to analyze the internal dynamics of the system. They determine the values and eigenvectors, which are used to analyze the dominance of existing loops, and try to explain their behavior through an analytical tool.

It should be noted that the authors only consider the first term of Equation (1) in the reduction of the system dynamics model to a linearized system invariant in time, making incomplete analysis of the linearized model and reducing the analysis only to the internal dynamics of the $\dot{X} = Ax$ system. In addition, the technique used to analyze the internal dynamics of the system, based on the matrix of partial derivatives in control engineering theory, has the objective of defining the internal stability of the system through the decomposition and interpretation of its characteristic canonical modes, which may belong to one of the following combinations: controllable and observable CO; not controllable and observable $\overline{C}O$; controllable and not observable $C\overline{O}$; and not controllable and not observable $\overline{C}\,\overline{O}$, which are not mentioned, or discussed, in the above researches. This presents an area of opportunity to perform a complete analysis of the systems dynamics models, and to strengthen the theory that supports the explanation of the system structure link-dynamic behavior for decision making.

The reason for this research, is that by proposing a methodology based on conventional control engineering theory, we can determine a transfer function of the SD model based on the Laplace transform. This explains the setting state variable trajectories to impulse or system inputs, transforming the time domain differential equations for solution to a frequency domain, and analyzing the external stability and controllability of the system based on the decomposition of controllable characteristic modes (poles) and observable (zeros) in the polynomial of the transfer function. Additionally, it aims to control and manipulate the behaviors of the model's significant variables and more robust decision-making, in contrast to the research cited where only the dominance of feedback loops is analyzed. Further, the analysis of the internal and external stability of the system, based on the decomposition and interpretation of characteristic modes (eigenvalues) is not contemplated, or the determination of a function that explains the particular trajectories to predict the response of the system to changes in the input variables, which as mentioned above, is not concluded by not interpreting its result to determine whether the system is stable or not stable, and not to generate the particular trajectory of each state variable as a system response.

One of the applications of great relevance for the proposed research is that generated by Goncalves [34]. Its main contribution is the determination of a function in time that explains the internal dynamics of the particular trajectory of each state variable, instead of just analyzing the internal dynamics with corresponding eigenvalues. The trajectory of the internal dynamics of the state variable laying consists of a linear combination of the product of the components of the eigenvector as coefficients, and the modes of behavior generated by eigenvalues as a response to the internal dynamics of the system. The contribution concludes that the derivatives of the own vectors are associated with the impact of the transients in the short term, and the own values are associated with the impact in the long term. It should be noted that the author uses the same technique of reducing the model to an invariant in the time of spaces of states, and in the same way that the other authors mentioned only using the first term of Equation (1) to perform the analysis of the internal stability of the system, and do not conclude defining whether there is stability or instability based on behavior through the calculation of values and eigenvectors of the partial derivative matrix.

The author finds a function to analyze the internal dynamics of the Ax system, making a transformation of coordinates of the original matrix to the corresponding products of the eigenvector with the corresponding eigenvalue. However, in the proposed function to analyze the internal dynamics of the system, it is not able to explain the behavior of the response to the internal dynamics and, likewise, it is not able to respond to the inputs or impulses of the system since they are not related in the equation found. We can also note that by analyzing only the decomposition of the characteristic modes of controllable

and observable, we thus contribute a methodology that serves as scientific support in the mathematical formality of SD models as a mechanism to explain the structure of the system based on dynamic behavior.

A manufacturing system consists of the transformation of raw material through a sequence of n operations. Each of the n operations can be represented in a system dynamics model through state variables. Each of the operations must meet a specific goal according to the corresponding order, which represents the entry to the system. This feature allows you to establish a balancing loop in each state variable, to analyze the fulfillment of these goals as the system's response to the input. Each of the n state variables will be analyzed as n SD Forrester diagrams, according to the sequence order of the manufacturing system. Each n SD Forrester diagram is transformed into a block diagram, doing the cascade reduction of the system, and obtaining the transfer function of each n Forrester diagram representing each of the state variables. In obtaining the transfer equations, an explanation of the behavior of the response as its particular trajectories to its corresponding inputs is made, and the stability, observability, and external controllability for each of them are analyzed.

The case study selected to validate the proposed methodology is the knitting textile manufacturing system for making children's sweaters. The process consists of seven operations, which represent the seven state variables involved in the SD simulation models. Entries to each of the operations are the established production goals. Seven SD simulation models were defined, each representing the sequence in the order of the textile process.

## 2. Materials and Methods

The case study to implement the proposed methodology is the knitting textile process, which consists of seven process operations. The main parameters to model each of these is their respective cycle times; that is, the time spent in making a sweater. See Table 1. Each of the different departments represent a state variable. The establishment time is determined for a production of 72 sweaters, considering the number of machines available in each process operation.

**Table 1.** Cycle times of the textile manufacturing process.

| State Variable | Cycle Time (minutes) | Number of Machines | Establishment Time for 72 Sweaters |
|---|---|---|---|
| Knitting | 40 | 6 | 480 |
| Basting | 3 | 2 | 108 |
| Ironing | 4 | 2 | 144 |
| Cutting | 3 | 4 | 54 |
| Making | 7 | 6 | 84 |
| Finishing | 2 | 2 | 72 |
| Packing | 2 | 2 | 72 |

The notations of the state variables, and representation of the differential equations, can be represented in Table 2, which are useful and necessary for Forrester diagrams and block diagrams. From a sensitivity analysis, the k parameters of each differential equation were determined. The values of k are based on the accumulated cycle times, considering the number of machines in each of them. Each process is simulated in a general way with an added cycle time in order not to perform the simulation on each of the machines of each operation.

The level of production in each operation gives rise to a first order differential equation. If n(t) denotes the number of sweaters produced in the state variable at time t, then the speed at which n(t) changes is a net speed:

$$dn/dt = (\text{input rate}) - (\text{output rate})$$

*Methodology*

The methodology was carried out in eight steps, which can be seen below, for the realization of this research project. See Figure 1.

**Table 2.** Data on auxiliary variables and notation of state variables.

| State Variable (m) | k | Notation of the Variable | Differential Equations |
|---|---|---|---|
| Knitting | 0.009541 | a | $da/dt = k_1(Xd_1 - a)$ |
| Basting | 0.12 | b | $db/dt = k_2(Xd_2 - b) - k_1(Xd_1 - a)$ |
| Ironing | 0.12 | c | $dc/dt = k_3(Xd_3 - c) - k_2(Xd_2 - b)$ |
| Cutting | 0.45 | d | $dd/dt = k_4(Xd_4 - d) - k_3(Xd_3 - c)$ |
| Making | 0.45 | e | $de/dt = k_5(Xd_5 - e) - k_4(Xd_4 - d)$ |
| Finishing | 0.5 | f | $df/dt = k_6(Xd_6 - f) - k_5(Xd_5 - e)$ |
| Packing | 0.55 | g | $dg/dt = k_7(Xd_7 - g) - k_6(Xd_6 - f)$ |

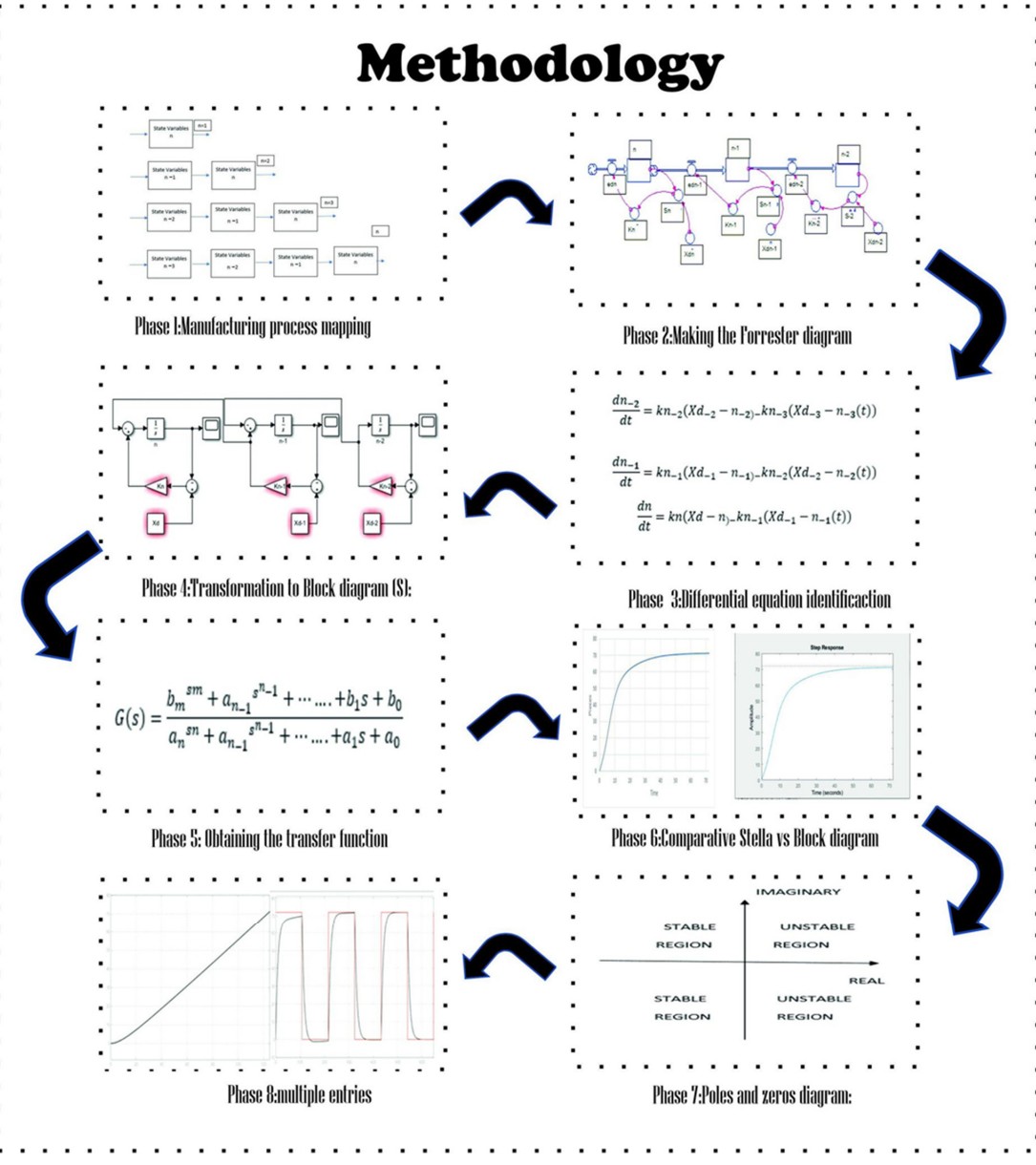

**Figure 1.** System dynamics methodology. Manufacturing process mapping: a mapping is made of the operations involved in the manufacturing process studied and carried out ascendingly until reaching

the n operations. Making the Forrester diagram: the Forrester diagram is made in the Stella software, where the state or stock variables, flow variables, feedbacks, and auxiliary variables involved in the manufacturing process are identified. Differential equation identificaction: the differential equations of the simulation model are identified, from the Forrester model; the differential equations are those that model mathematically and serve as support for the simulation model. Transformation to Block diagram (S): the transformation of the Forrester diagram to the block diagram is carried out by means of the differential equations that model the simulation system. Obtaining the transfer function: two main equations are proposed, the direct path and the feedback diagram; with the help of the MATLAB and Simulink software the transfer function for the study variable is obtained. Comparative Stella vs. Block diagram: the transfer function is plotted with the system input and compared with the graphics obtained by the Stella software and the same goal or input, to compare whether the transformation performed is correct. Poles and zeros diagram: with the transfer function of the system, the equations characteristic of the numerator and denominator are analyzed to find its roots, where the roots found in the numerator represent the zeros and the roots found in the denominator represents the poles or eigenvalues of the system; if the eigenvalues are in the left semiplane of the pole and zero diagram, this means that the process with the n variables under study is stable, otherwise it is unstable. Multiple entries: the behavior of the transfer function obtained in each of the models, before inputs type pulse train and ramp type.

## 3. Results

### *3.1. Proposed Approach to the Knitting Operation*

#### 3.1.1. Manufacturing Process Mapping

The n operations involved in the manufacturing system are mapped in such a way that the process of conceptualization for each of the state variables is carried out in stages; from 1 to the n state variables that the process of manufacture has, as seen in Figure 2.

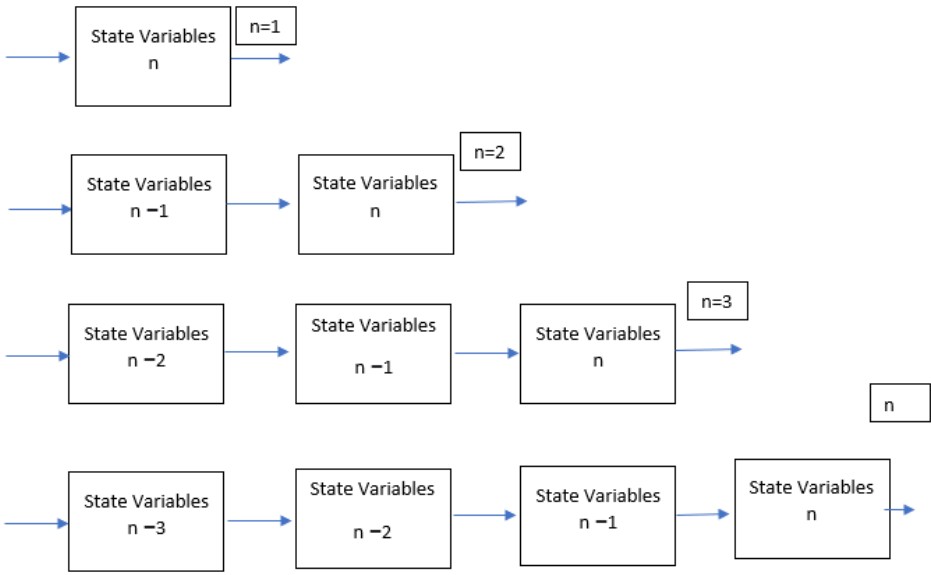

**Figure 2.** Mapping diagram of n manufacturing operations.

#### 3.1.2. Forrester Diagram and Identification of the Differential Equation

When a system consists of multiple negative loops, any action that attempts to modify an element is not countered by the loop in which the element is located, but by the whole set of negative loops that act in its support, by super-stabilizing the system. An analysis of the system can help us in any complex system, whether social or industrial, and is made up of hundreds of elements. Each element relates only to a limited number of variables that are important to it, and that it permanently compares to its objectives. If there is a discrepancy between the status of these variables and their objectives, the element acts in a certain way to modify the system. The greater this discrepancy, the greater the action

carried out by the element in the system, in order that all the variables of state of the process reach their objective; in the case of this study, there are 72 input parts of the system. The Forrester diagram is made with seven state variables and seven negative balancing loops; in the textile process in the Stella software, the trajectories that explain the behavior of the variables of state of the textile process, are related. You can see the textile model from right to left, because the filling of the flow variables in Stella is given from right to left and is explained mathematically in the methodology, as the order of the n state variables. The SD model can be seen in Figure 3.

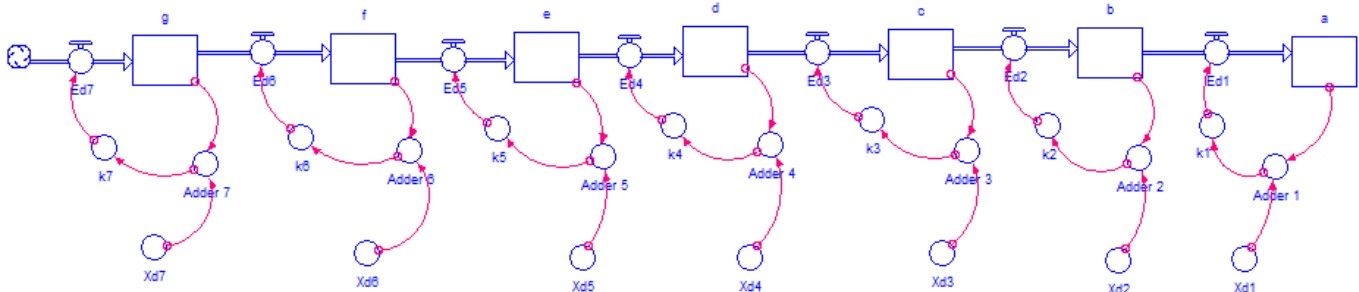

**Figure 3.** Forrester diagram of the textile process.

The identification of each of the state variables that are related in the manufacturing process is performed, starting with one variable state for the knitting process. The DS loop used in the Forrester diagram of this application is the rolling loop, which seeks to evoke the behavior of looking for a goal. See Figure 4.

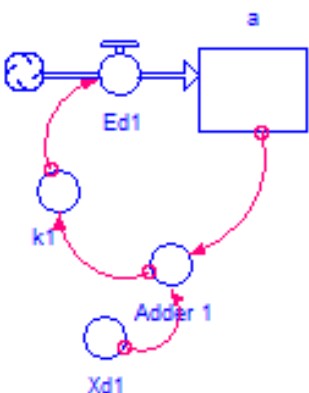

**Figure 4.** Forrester diagram for n = 1 variable state.

The differential equation of the system is identified from the Forrester diagram, $Xd_1$ is the goal of the system, a(t) function at the time of the process, while k is the value that makes it so fast that you reach the goal of the system, where you have the first balancing loop. The goal is subtracted with the output observed in the system, and the difference between them is multiplied by the time constant, which are equal to the differential equation or variable flow of the system. See Equation (3).

$$\frac{da}{dt} = k_1(Xd_1 - a), \tag{3}$$

It should be noted that the transient characteristic is known from Forrester simulation models, based on the system dynamics methodology. This transient characteristic is calculated by numerical methods by simulation in the Stella program, for example the Euler and Newton Raphson method.

Later, with the proposed research, the Forrester diagram is transformed into block diagrams to obtain the respective transfer function, and it is checked by the poles and zeros obtained by the transfer function that the system is stable.

### 3.1.3. Knitting Process Block Diagram

The transformation of Forrester diagram to block diagram is carried out, starting from the identification of the differential equation that model the Forrester diagram. The transformation in the Laplace domain is performed for n = 1. The value of k works as to how quickly the system reaches the goal. The model of the block diagram is as follows. The system for a state variable functions, when performing the goal comparison of the system minus the output of the same, that difference between the real and the desired, is multiplied by the k; or that factor that performs the period-to-period adjustment and makes the comparison of the real with the desired, until the real is equal to the desired goal and the error is 0. See Figure 5.

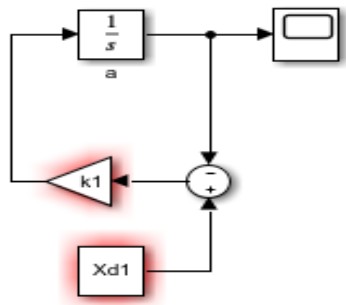

**Figure 5.** Diagram of blocks of 1, state variables.

For the reduction of the block diagram, the MATLAB Simulink tool is used, where the design of the block diagram is performed and the parameters of k and $Xd_1$ are introduced, thus obtaining the transfer function of the system in Equation (4).

$$H_a(s) = \frac{1}{(Ts + 1)} \tag{4}$$

where

$$T = 1/0.009541$$

From the transfer function, with the help of the MATLAB and Simulink software by means of the oscilloscope, the output can be observed before a step input with an amplitude of 72. A comparison of the result obtained with MATLAB and Stella can be seen in Figure 6.

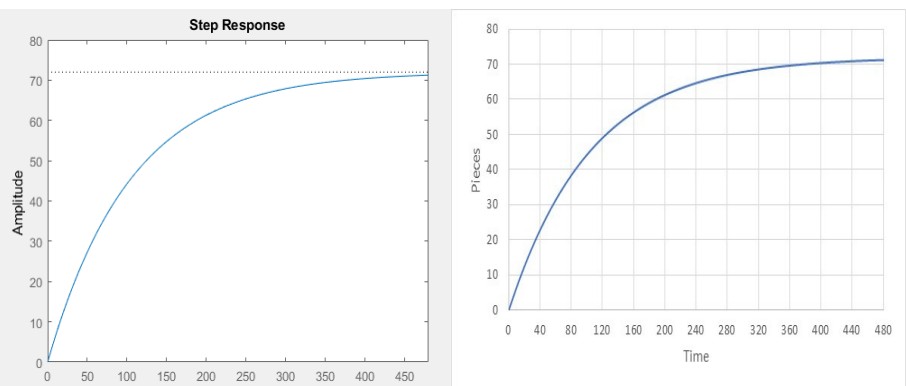

**Figure 6.** Output process of weaving, blocks and Stella.

Where you have to compare the values obtained by the system output with an input of 72, or goal of 72 in Stella, you have an average quadratic error of less than 0.0001%, which comes from the number of decimals handled by the Stella software.

### 3.1.4. Stability of the Knitting Process System

For the first order system represented by the transfer function obtained, a positive coefficient is obtained, which indicates that the system is stable, and by equalizing the denominator of the function, a real pole located in the negative left semi-plane is obtained. See Figure 7.

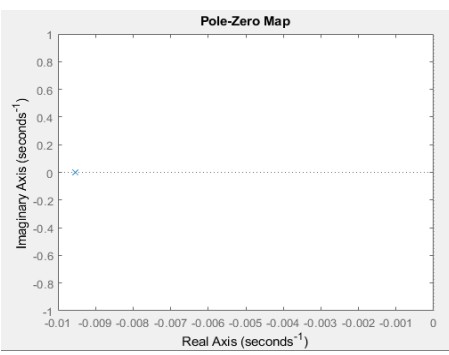

**Figure 7.** Diagram of poles and zeros weaving process.

The establishment time to reach the target by 98% consists of four times the value of T. For the system, the establishment time is 419.28 min. To get the value in the stationary state you do s = 0 and you get that it is one, for the first order mathematical function.

### 3.1.5. Multiple Inputs-Knitting Process

By obtaining the system transfer function, one of the great advantages of having the mathematical model is to obtain the graphical solution of multiple external inputs to the system; whereas by solving by differential equations, one has the mathematical function for an input with a specific amplitude. Next, the behavior of the output of the knitting process is visualized, with a ramp type input that simulates the behavior of the process, and an impulse generator that simulates dead times of 50% of productivity, having as orders the 72 orders in several occasions. See Figure 8.

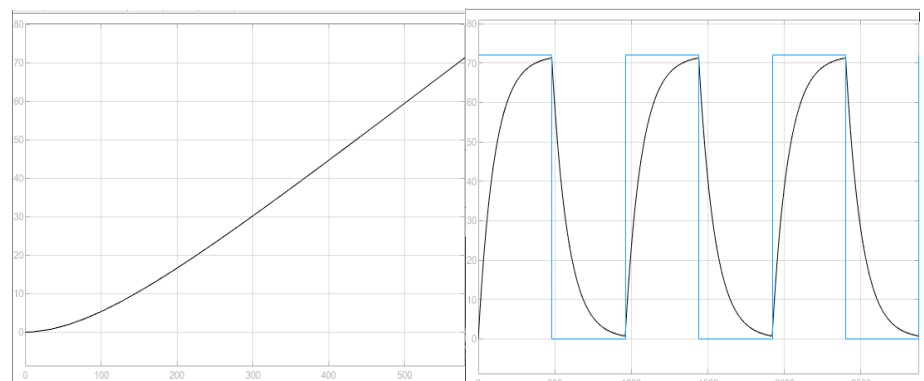

**Figure 8.** Response of the knitting process to a ramp and pulse generator input per cycle, n = 1.

For a ramp input, the process behavior remains stable, but the set time is increased by about 560 min, because the target increases linearly with a slope of 0.15. For a pulse input with a period of 960, the step is activated 50%, the target is reached in approximately 480 min.

### 3.2. Proposed Approach to the Basting Operation

### 3.2.1. Forrester Diagram-Basting Process

The Forrester diagram is presented for two state variables, where there are six auxiliary variables, two flow variables and two stock or state variables. You have two rolling loops,

variable a, not affected by the behavior of b, while variable b is affected by the state variable a directly, as long as a reaches its goal faster. See Figure 9.

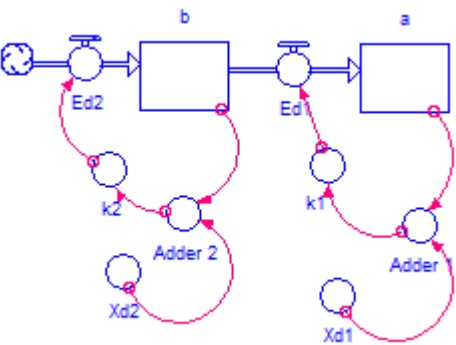

**Figure 9.** Forrester diagram for the basting process with two state variables.

Next, the two differential equations are identified for the construction of the block diagram and we get the respective transfer function. It should be noted that the net rates of production are determined in a simple way, such as the input rate minus the output rate. The first differential equation was determined in the previous operation as Equation (3). The rate of velocity is determined for state variable two in Equation (4).

$$\frac{db}{dt} = k_2(Xd_2 - b) - k_1(Xd_1 - a) \tag{5}$$

### 3.2.2. Diagram of Blocks-Basting Process

It is proposed to solve the system of differential equations through the block diagram technique, which is a control engineering technique. The Simulink software is used to obtain the transfer function, as seen in Figure 10. The two state variables are in the block diagram, where clearly the state variable b also depends on the flow of a.

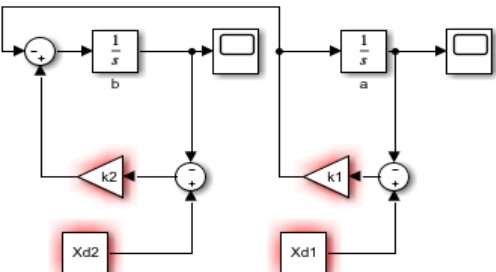

**Figure 10.** Block diagram for basting process with two state variables.

For the solution of two state variables, it is very useful to use MATLAB Simulink tooling, where the transfer function system poles are equal to the number of state variables, then the system transfer function is presented in Equation (6).

$$H_b(s) = \frac{0.1104s + 0.001145}{s^2 + 0.1295 + 0.01145} \tag{6}$$

From the transfer function obtained, and with a step input with an amplitude of 72, the behavior of the basting process is simulated and compared with the simulation obtained from the Forrester diagram. See Figure 11.

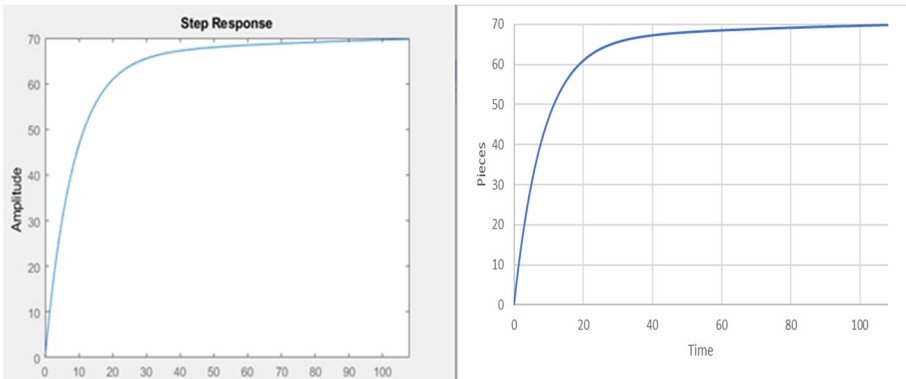

**Figure 11.** Output process of basting.

When comparing the values obtained by the system output with an input of 72, or goal of 72 in Stella, identical outputs are obtained, which validate the transformation made from SD to the block diagram based on control engineering.

### 3.2.3. Stability of the Basting-System-Process

We have the characteristic equations of the numerator and denominator of the transfer function of two state variables, from the diagram of poles and zeros for n = 2.

The poles or eigenvalues = $-0.12$ and $-0.0095$

The zeros = $-0.0104$

Two real poles and one real zero are obtained, where the representation in the pole-zero diagram can be seen in Figure 12.

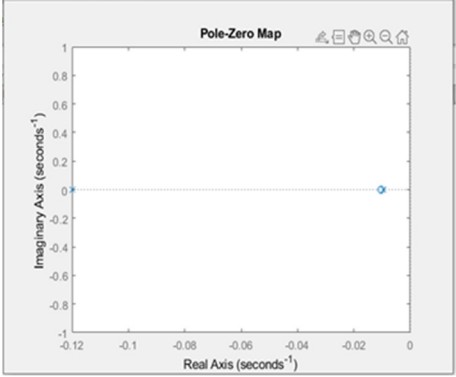

**Figure 12.** Pole zero diagram, two state variables.

The resulting poles in the left half-plane of the diagram are on the real line, which confirms the non-surge behavior of the output to a step input with an amplitude of 72. It can be concluded that, given the eigenvalues of the system, for two state variables it is stable.

### 3.2.4. Multiple Inputs-Basting Process

The transfer function obtained for two state variables is different in the degree of the characteristic polynomial, but it has in common that it is able to observe the output of the system for multiple inputs, regardless of whether the degree of the transfer function is greater. This is one of the great advantages of obtaining the transfer function; the response to a ramp input that simulates the tracking of the goal is obtained, in addition to an input type pulse generator with an amplitude of 72 in 50% of the cycle, and a period of 216 min. See Figure 13.

The clearance, or follow-up time, to reach the goal with a ramp entry is about 10% longer, which is logical because the process is followed, and there are delays between

real and recorded. The step input in its first cycle has a faster behavior than the previous state variable, but when modeling two or more cycles the transient response is even faster, because we take into account that it is not less affected by the previous state variable. See Figure 13.

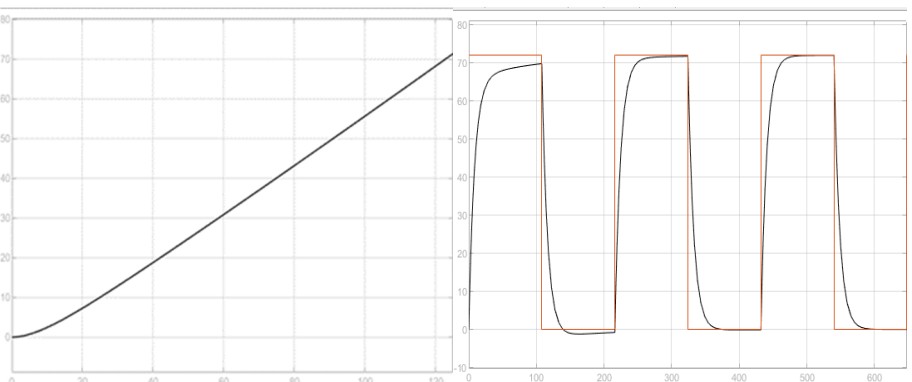

**Figure 13.** Response of the spinning process to a ramp and step input per cycle.

For the second and third cycle, the establishment time is faster than the first cycle. The process has initial conditions of 0 and does not have the influence of the knitting process. The transfer function that governs the behavior of any cycle greater than 1 is in Equation (7).

$$H_a(s) = \frac{1}{\left(\frac{s}{0.12} + 1\right)} \tag{7}$$

where the constant time T of the system transfer function is 8.333. The establishment time to reach 98% of the target is 33.33 min.

### 3.3. Proposed Approach to the Ironing Operation

3.3.1. Forrester Diagram-Ironing Process

The Forrester diagram is presented for three state variables, where there are nine auxiliary variables, three flow variables and three stock or state variables. One of the big differences with three state variables rather than with two, is that the state variable c, is directly limited by the state variable b, but this in turn is limited by the state variable a. Therefore, the mathematical model becomes more robust; it would need a system of three differential equations to obtain its output for a given input, i.e., the solution of b and a is already needed to obtain c. See Figure 14.

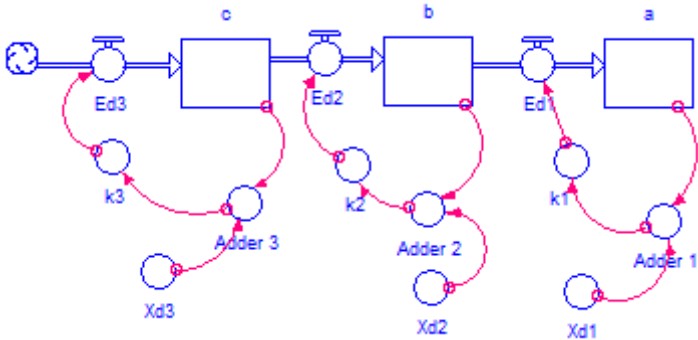

**Figure 14.** Forrester diagram for the ironing process with three state variables.

Then, the three differential equations are identified for the construction of the block diagram and to get the respective transfer function. The net rates of production are determined by subtracting the rate of entry minus the rate of production. The first and second

differential equation were determined in previous operations, in Equations (3) and (5), respectively. The rate of velocity is determined for the state variable three in Equation (8).

$$\frac{dc}{dt} = k_3(Xd_3 - c) - k_2(Xd_2 - b(t)) \tag{8}$$

### 3.3.2. Block Diagram-Ironing Process

The system of differential equations is solved through the block diagram technique, which is a control engineering technique. The MATLAB Simulink tool is used to get the transfer function. As can be seen in Figure 15, in the three state variables in the block diagram you see the state variable c, where it also depends on the flow of b and a.

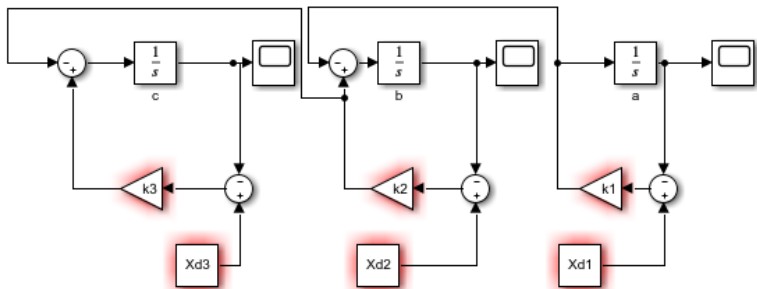

**Figure 15.** Block diagram of n = 3 ironing state variables.

With the help of the MATLAB and Simulink software we get the system transfer function for n = 3. Since the values of $k_2$ and $k_3$ are identical, a slight increase in the value $k_3$ was made to avoid the cancellation of a zero in the numerator polynomial. The transfer function obtained is determined in Equation (9).

$$H_c(s) = \frac{0.001s^2 + 01338s + 0.0001385}{s^3 + 0.2505s^2 + 0.01682s + 0.0001385} \tag{9}$$

From the transfer function, and with a step input with an amplitude of 72 and a target of 72, the following plots are obtained, which reach the target in approximately 144 min. See Figure 16.

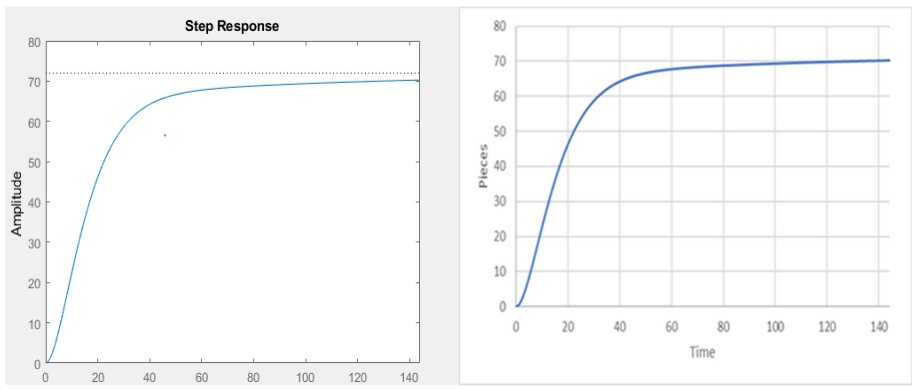

**Figure 16.** Ironing process output at step input and Stella output.

### 3.3.3. System Stability-Ironing Process

The transfer function of the numerator is 2°, while the denominator is 3°, the k's are different between the state variable b and c, so you have an n − 1 degree in the numerator with respect to the denominator.

The eigenvalues and eigenvectors of the transfer function are:

Poles = −0.121, −0.12 and −0.0095

Zeros = −0.0104

There are three real poles, of which two real poles are repeated and one real zero. See Figure 17.

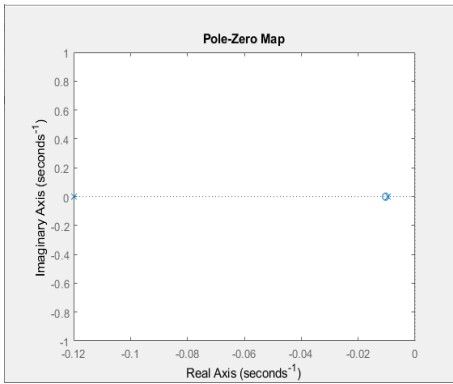

**Figure 17.** Diagram of poles and zeros of three state variables.

The resulting poles in the left half-plane of the diagram, which can be concluded for n = 3, is stable for the poles and zeros found, with the poles of the system being real, a stable behavior is expected in the case of multiple inputs to the system.

### 3.3.4. Multiple Inputs-Ironing Process

It has a ramp input with a slope of 0.5 in 144 s and an impulse generator with a period of 288 and an amplitude of 72 in 50% of the cycle. See Figure 18.

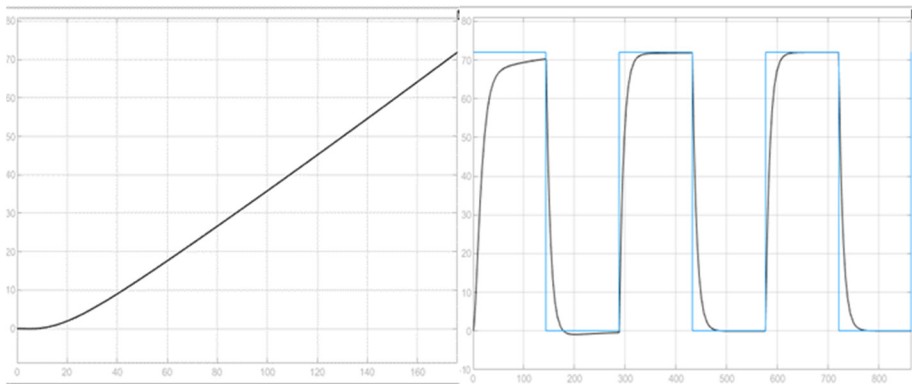

**Figure 18.** Response of the plate process to a ramp and pulse generator input by cycles, n = 3.

The preparation time is 176 min, about 15% longer than the time needed to reach the target; the delay increases due to the delay of the previous departments by about 5% with the immediately previous one. The transient response is slower than in the previous process, you have an establishment time of approximately 144 min.

### 3.4. Proposed Approach to the Cutting Operation

### 3.4.1. Forrester Diagram-Cutting Process

The Forrester diagram is presented for four state variables, where there are 12 auxiliary variables, four flow variables and four stock or state variables. The state variable d, is directly limited by the state variable c, but this in turn is limited by the state variable b, and finally this in turn is limited by the state variable a. Therefore, the mathematical model becomes more robust; it would need a system of four differential equations to obtain its output. See Figure 19.

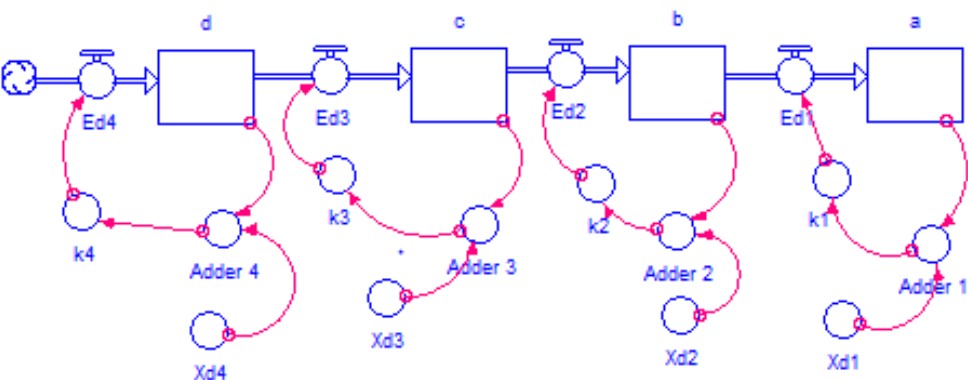

**Figure 19.** Forrester diagram for the cutting process with n = 4.

Then, the four differential equations are identified for the construction of the block diagram and to get the respective transfer function. The net rates of production are determined by subtracting the rate of entry minus the rate of production. The first, second and third differential equation were determined in the previous operations in Equations (3), (5) and (7), respectively. The rate of velocity is determined for the state variable four in Equation (10).

$$\frac{dd}{dt} = k_4(Xd_4 - d) - k_3(Xd_3 - c) \tag{10}$$

### 3.4.2. Block Diagram-Cutting Process

It is proposed to solve the system of differential equations through the block diagram technique, which is a technique based on control engineering. The Simulink simulation tool from MATLAB is used to obtain the transfer function. As can be seen in Figure 20, there are four state variables in the block diagram, where clearly the state variable d also depends on the flow of the three previous state variables. See Figure 20.

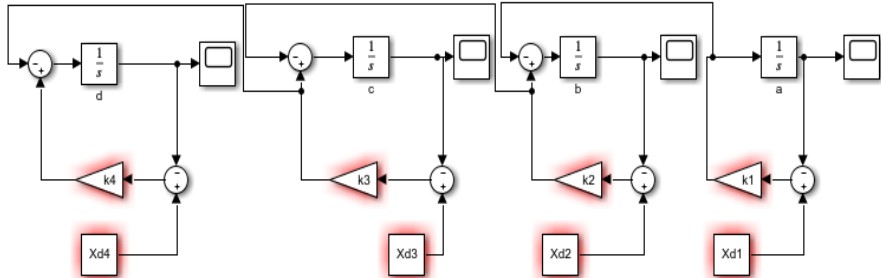

**Figure 20.** Block diagram for the cutting process, with n = 4.

With the help of the MATLAB and Simulink software, the transfer function of the system is obtained by designing the block diagram in cascade and replacing the values of $k_1$, $k_2$, $k_3$, $k_4$, $Xd_1$, $Xd_2$, $Xd_3$ and $Xd_4$, to obtain the transfer function in Equation (11).

$$H_d(s) = \frac{0.33s^3 + 0.08235s^2 + 0.007098s + 6.182 \times 10^{-5}}{s^4 + 0.6995s^3 + 0.129s^2 + 0.007648s + 6.183 \times 10^{-5}} \tag{11}$$

The transfer function of the numerator is $3°$, while the denominator is $4°$, the k's are different between the state variable c and d, so you have a n − 1 degree in the numerator with respect to the denominator. See Figure 21.

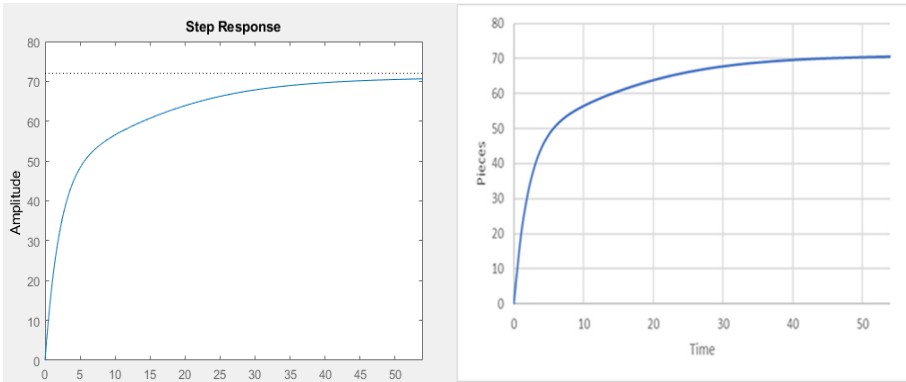

**Figure 21.** Cutting process output with a step input and Stella software.

When comparing the graphs obtained through the system output with an input of 72, the approximate time to obtain the target in Stella and block diagram is 54 min.

### 3.4.3. Stability of the System-Ironing Process

The characteristic equations of the numerator and denominator of the transfer function found that when the poles and zeros are analyzed, four poles and three zeros will be obtained.

The eigenvalues and eigenvectors of the transfer function are:

Poles = $-0.45, -0.12, -0.12, -0.0095$

Zeros = $-0.1199 + 0.6921i, -0.1199 - 0.0629i$, and $-0.0098$.

There are four real poles and two conjugate imaginary zeros, and a real zero, resulting in the poles in the left half plane of the diagram, which can be concluded for a n = 4 is stable for the poles and zeros found; there are no oscillations in the system because the conjugate roots are of the zeros and not of the poles, that directly influence the behavior of the system output. See Figure 22.

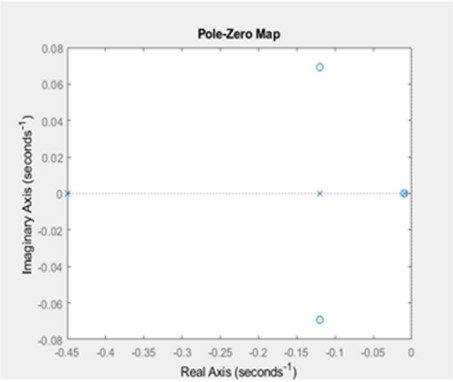

**Figure 22.** Plot of poles and zeros of n = 4, state variables.

### 3.4.4. Multiple Inputs-Cutting Process

For the cutting process, the transfer function obtained with a stepped pulse input with an amplitude of 72 in 50% of the cycle and zero in the other half of the period is tested, with a ramp type input with a linear slope of 1.333 pieces per minute. See Figure 23.

The cutting process has a faster response when reaching the 72 pieces as the objective of the manufacturing process. With different inputs, the process will be faster with respect to the previous processes; for multiple inputs, a setup time of 57 min is obtained, that is to say, besides being the process with the fastest response of all before a step input, it also takes approximately 5% to follow up the process. There is a quick exit response for a ramp entry with a linear slope of 1333 pieces per minute, the response is 57 min and the step entry is the fastest transient of all transfer functions presented, in about 54 min.

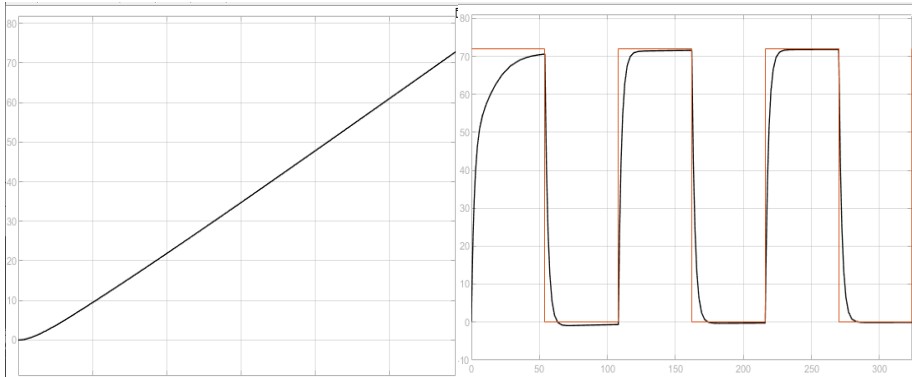

**Figure 23.** Response of the cutting process to a ramp input and a pulse generator n = 4.

### 3.5. Proposed Approach to the Making Operation

#### 3.5.1. Forrester's Diagram-Making Operation Process

There are 15 auxiliary variables, five flow variables and five stocks. The complexity of obtaining a 5th state variable is not even manageable and needs to be done by means of specialized software due to the number of differential equations to be performed. In the MATLAB software, to determine the function in time of the differential equation for the case study, it did not solve the solution in a suitable way as a function of time. On the other hand, there are five balancing loops that for the 5th state variable is affected by the whole system and there is a swift decrease of the process in the non-operation time of the process. See Figure 24.

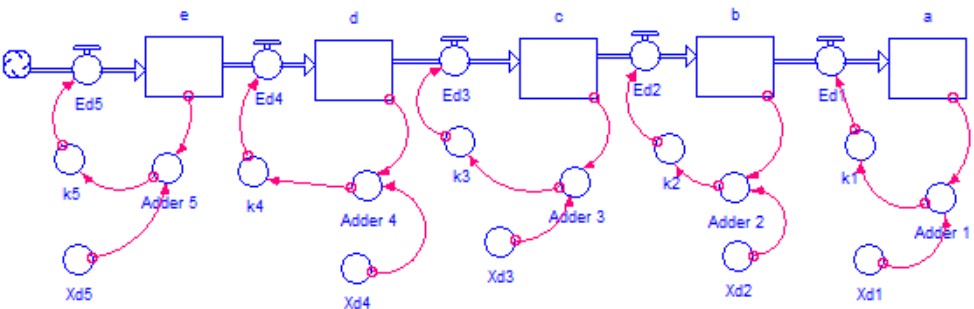

**Figure 24.** Forrester diagram of the confection process, n = 5.

The five differential equations are identified for the construction of the block diagram and obtain the respective transfer function. The first, second, third and fourth differential equations were determined in previous operations in Equations (3), (5), (7) and (9), respectively. The velocity is determined for the state variable five in Equation (12).

$$\frac{de}{dt} = k_5(Xd_5 - e) - k_4(Xd_4 - d(t)) \tag{12}$$

#### 3.5.2. Block Diagram-Making Operation Process

It is proposed to solve the system of five differential equations through the block diagram technique, which is a technique based on control engineering. MATLAB's Simulink simulation tool is used to obtain the transfer function. As can be seen in Figure 25, there are five state variables in the block diagram, where you see that the state variable e also depends on the flow of the four previous state variables.

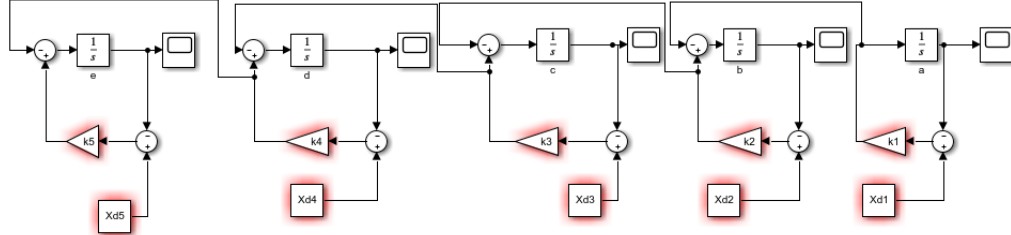

**Figure 25.** Block diagram for the confection process, with n = 5.

With the help of the MATLAB and Simulink software we get the system transfer function for n = 5. Since the values of $k_4$ and $k_5$ are identical, a slight increase in the value $k_5$ was made to avoid the cancellation of a zero in the numerator polynomial. The transfer function obtained is determined in Equation (13).

$$H_e(s) = \frac{0.001s^4 + 0.1488s^3 + 0.03728s^2 + 0.003221s + 2.805 \times 10^{-5}}{s^5 + 1.151s^4 + 0.4448s^3 + 0.06661s^2 + 0.003531s + 2.805 \times 10^{-5}} \tag{13}$$

where the step input defined by 72 parts is defined in the process and plotted for Stella and block diagram, which is represented by the system goal.

It is found that the settling time of the two plots is approximately 84 and the difference of the mean square error of the data in the two results has a mean square error of less than 0.001%, where the error is practically the number of significant figures handled by the Stella software. See Figure 26.

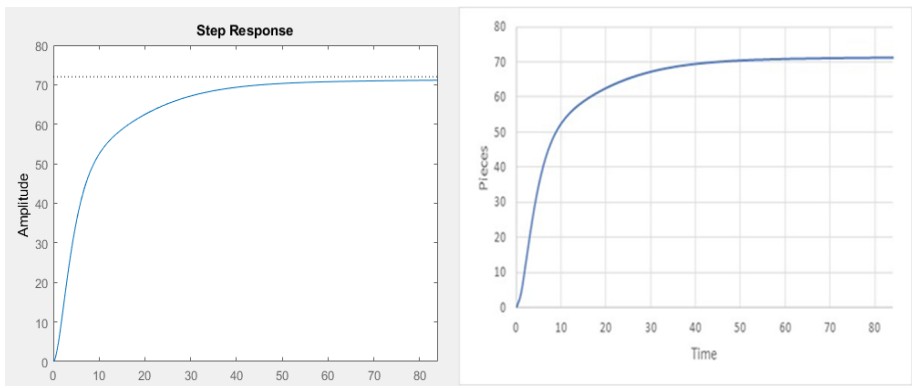

**Figure 26.** Output at a step input with amplitude 72 and with the Stella software.

3.5.3. Stability of the System-Making Operation Process

When the polynomial equals to zero and the roots of the numerator and denominator are taken out, there are three zeros and five poles, this is because the 4th and 5th state variable have the same k; that is to say, at the beginning of the process of the 5th state variable, it does not start the accumulation because initially what enters and leaves the flow variables ed5 and ed4 are the same.

Poles = $-0.451$, $-0.45$, $-0.12$, $-0.12$ and $-0.0095$

Zeros = $-0.1199 + 0.0692i$, $-0.1199 - 0.0692i$ and $-0.0098$

The resulting five poles on the real axis, two imaginary conjugate zeros and a zero on the real axis, are all in the left plane of the pole diagram, resulting in a stable system for the five state variables, where the poles of the system are equal to the value of the k, assigned in the Forrester diagram. See Figure 27.

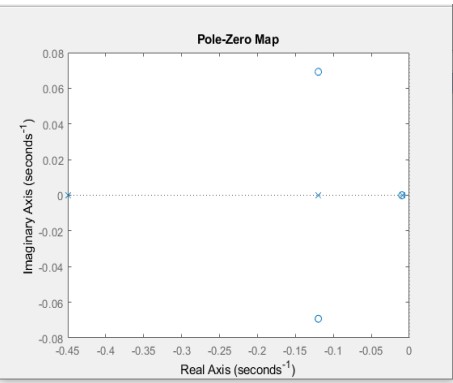

**Figure 27.** Plot of poles and zeros of n = 5, state variables.

### 3.5.4. Multiple Inputs-Making Operation Process

It has a ramp type entrance with a slope of 0.857 garments per minute on average and an impulse generator type input with a period of 168, which corresponds to double the delivery time and 50% inactivity.

It has a set time of 84 min approximately for the first half of the pulse train period and descends in the dead period, the ramp type setting time is 10% slower than the step. See Figure 28.

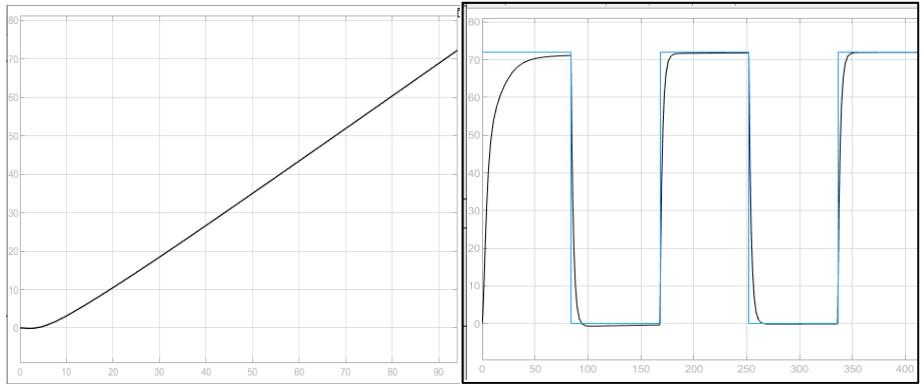

**Figure 28.** Response of the confection process to a ramp input and a pulse generator n = 5.

### 3.6. Proposed Approach to the Finishing Operation

### 3.6.1. Forrester Diagram-Finishing Operation Process

There are 18 auxiliary variables, six flow variables and six stocks. The complexity of obtaining a 6th state variable is not even able to be done by means of specialized software due to the number of differential equations to be performed. In the MATLAB software, to determine the function in the time of the differential equation for the case study, it did not solve the solution in a suitable way as a function of time. On the other hand, there are six balancing loops that for the 6th state variable is affected by the whole system; there is a fast decrease of the process in the non-operation time of the process. See Figure 29.

Then, the six differential equations are identified for the construction of the block diagram and obtain the respective transfer function. It should be noted that net rates of production are determined as the entry rate minus the production rate. The first five differential equation was determined in previous operations as Equations (3), (5), (7), (9) and (11). The rate of development is determined for the six states variable in Equation (14).

$$\frac{df}{dt} = k_6(Xd_6 - f) - k_5(Xd_5 - e(t)) \tag{14}$$

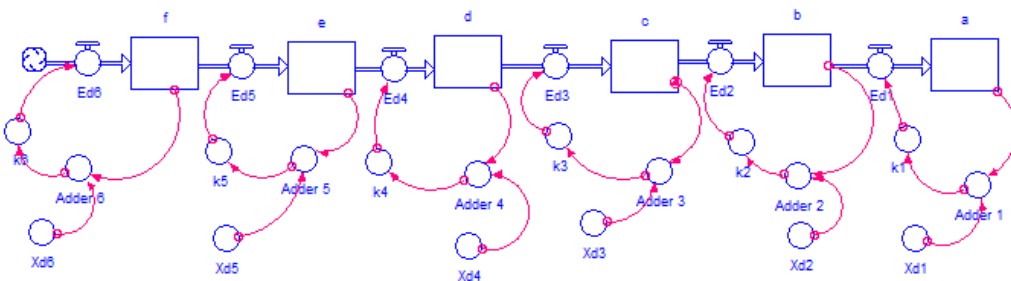

**Figure 29.** Forrester diagram of the finishing process with six state variables.

### 3.6.2. Block Diagram-Finishing Operation Process

It is proposed to solve the system of differential equations through the block diagram technique. It obtains the transfer function by means of Simulink. As can be seen in Figure 30, there are six state variables in the block diagram, where clearly the state variable f also depends on the flow of the five variables above.

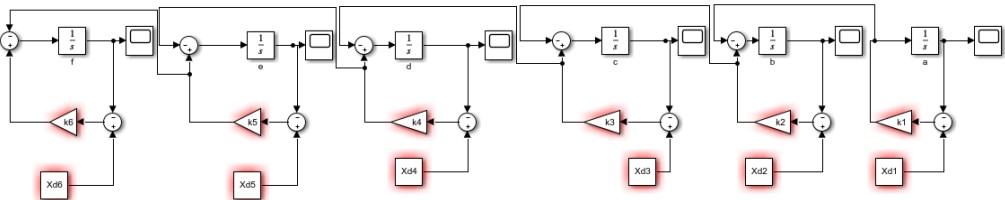

**Figure 30.** Block diagram of the finishing process, n = 6.

With the help of the MATLAB and Simulink software, we enter the parameters of the k and Xd for each of the six state variables, we point out the inputs and outputs of the system, and we obtain the transfer functions of the system in Equation (15), where we have a polynomial of degree 5° in the numerator and 6° in the denominator.

$$H_f(s) = \frac{0.05s^5 + 0.05748s^4 + 0.08901s^3 + 0.01996s^2 + 0.0016134s + 1.391 \times 10^{-5}}{s^6 + 1.65s^5 + 1.019s^4 + 0.2876s^3 + 0.03635s^2 + 0.001779s + 1.391 \times 10^{-5}} \quad (15)$$

By obtaining the transfer function we can see the behavior before multiple inputs, but the restriction or order of the case study is 72, and can be compared with the result obtained in the Stella software.

It is found that the establishment time of the two plots is approximately 72 min and the difference is in decimals. See Figure 31.

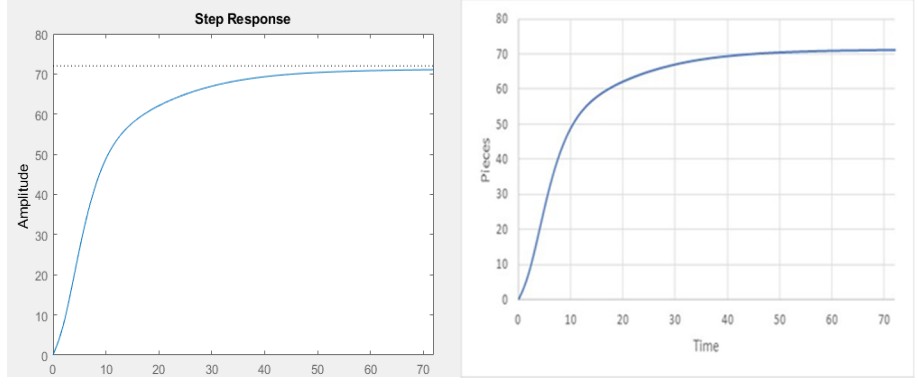

**Figure 31.** Output at a step input of 72 and the Stella software.

### 3.6.3. System Stability-Finishing Process

For the finishing process, and with six state variables, the numerators and denominators are equalized to find their characteristic roots, which in this case have exactly one pole more than zeros.

Poles = $-0.5$, $-0.45$, $-0.45$, $-0.12$, $-0.12$ and $-0.0095$

Zeros = $-0.4510 + 1.1544i$, $-0.4510 - 1.1544i$, $0.1189 + 0.0666i$, $-0.1189 - 0.0666i$, and $0.0098$.

There are four conjugate imaginary zeros and a zero on the real axis. While the poles have six poles on the real axis, it is very important to pay attention in the processes of the pole of $-0.009541$, because it is very close to the imaginary axis and can enter the process in marginally stable. See Figure 32.

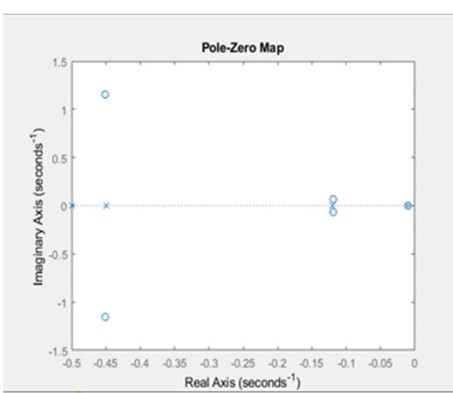

**Figure 32.** Pole-zero diagram of n = 6 state variables.

### 3.6.4. Multiple Inputs-Finishing Process

Ramp inputs and a pulse generator with a period of 144 min with 50% and with an amplitude of 72; the other interval is with 0 amplitude, and the ramp input is with an average of 1 piece per minute.

For a ramp entry you get a response time of approximately 80 min for the recording of the 72 pieces, while for a step type input you have a transient of 72 min and a speed increase in the second cycles of the process, because in the seconds or n cycles you take only one cycle of the previous processes. See Figure 33.

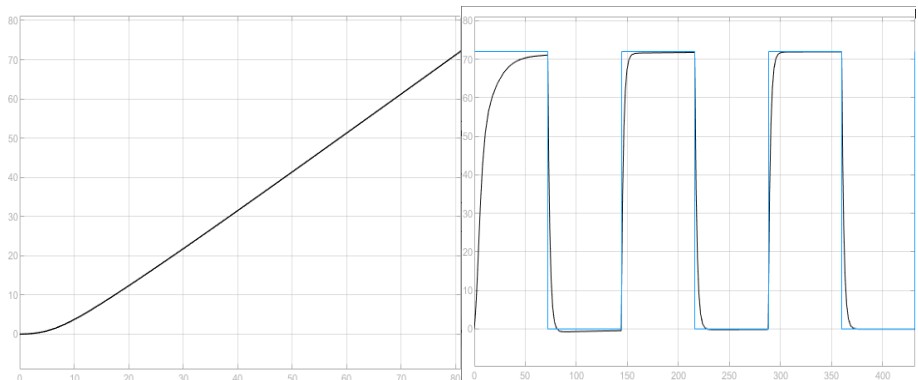

**Figure 33.** Response of the finished process to a ramp input and a pulse generator, n = 6.

### 3.7. Proposed Approach to the Packing Operation

### 3.7.1. Forrester Diagram-Packing Operation Process

The Forrester diagram is presented for seven state variables, where there are 21 auxiliary variables, seven flow variables and seven stock or state variables. For seven state variables, it is necessary to have the function in time of six previous state variables; basically it would be a system of linear differential equations, which must be solved in cascade. See Figure 34.

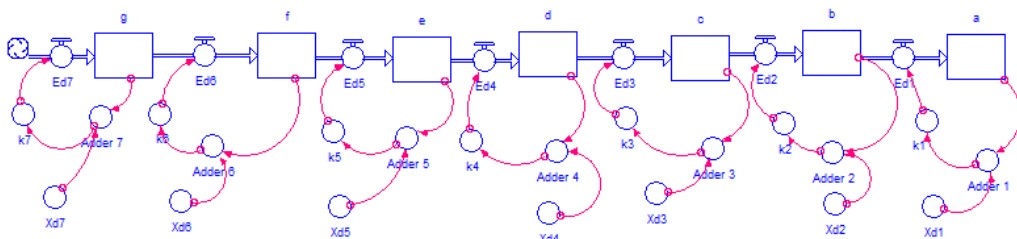

**Figure 34.** Forrester diagram of the packaging process, n = 7.

For the approach with seven state variables, it is identified that you must have seven differential equations for the formulation of the block diagram, with the Equations (3), (5), (7), (9), (11), (13) and Equation (16) corresponding to the flow of the state variable of the packing process.

$$\frac{dg}{dt} = k_7(Xd_7 - g) - k_6(Xd_6 - f(t)) \tag{16}$$

### 3.7.2. Block Diagram-Packing Process

It is proposed to solve the system of seven differential equations through the block diagram technique, which is a control engineering technique for obtaining the transfer function of the packing process. As can be seen in Figure 35, there are seven state variables in the block diagram, where clearly the state variable g also depends on the whole process.

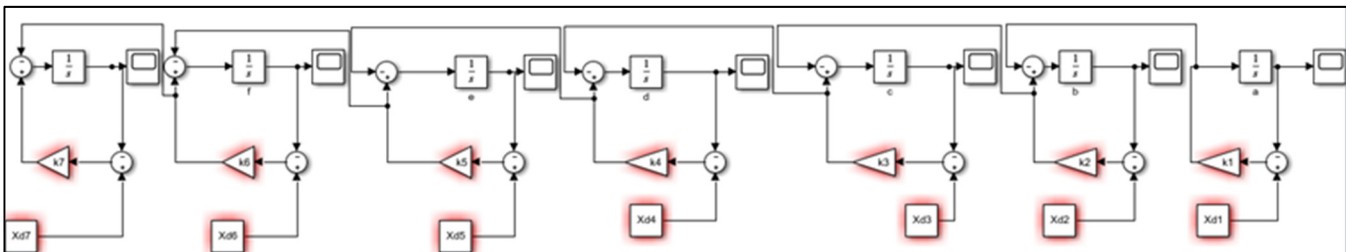

**Figure 35.** Block diagram of the packaging process, n = 7.

With the help of the MATLAB and Simulink software, the defined parameters are assigned to all 7 k and the goals of the system, which are 72 in each one of them. The transfer function of the system is obtained in Equation (17). We have a numerator of degree 6° and a denominator of degree 7.

$$H_g(s) = \frac{0.05s^6 + 0.1075s^5 + 0.0796^4 + 0.05889s^3 + 0.0118s^2 + 0.0008953s + 7.651 \times 10^{-5}}{s^7 + 2.2s^6 + 1.926s^5 + 0.8478s^4 + 0.1945s^3 + 0.02177s^2 + 0.0009926s + 7.651 \times 10^{-5}} \tag{17}$$

Obtaining the transfer function of the complete process, we can observe the output of the system, which is approximately 72 min for the packaging process; we have an analysis of the results in Stella with the transfer function. See Figure 36.

### 3.7.3. Stability of the System-Packing Process

The poles and zeros of the transfer function are analyzed for seven state variables, the seven poles and six zeros of the process are found, which can be seen below.

Poles = $-0.55, -0.50, -0.45, -0.45, -0.12, -0.12$ and $-0.0095$

Zeros = $-0.1760 + 0.72651i, -0.1760 - 0.72651i, -0.1180 + 0.0649i, -0.1180 - 0.0649i,$ $-1.5518$ and $-0.0097$.

Figure 37 shows seven poles on the real axis, where there is a behavior without over peak or oscillations, by having all the poles of the system on the real axis; additionally, four imaginary and two real conjugate zeros were obtained.

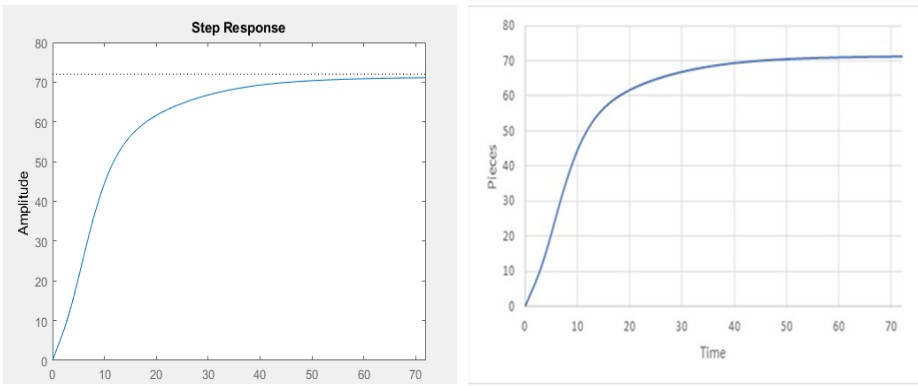

**Figure 36.** Output before a step input with an amplitude of 72, in Stella.

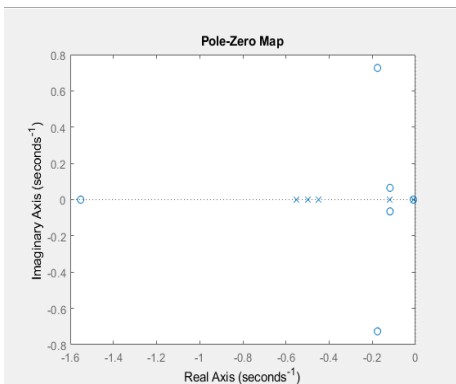

**Figure 37.** Plot of poles and zeros of seven state variables.

### 3.7.4. Multiple Inputs-Packing Process

The following are the ramp type inputs and pulse generator with a period of 144 min, where 50% of the input is 0 and the other 50% is 72, a ramp type input with a slope of exactly one garment per minute. See Figure 38.

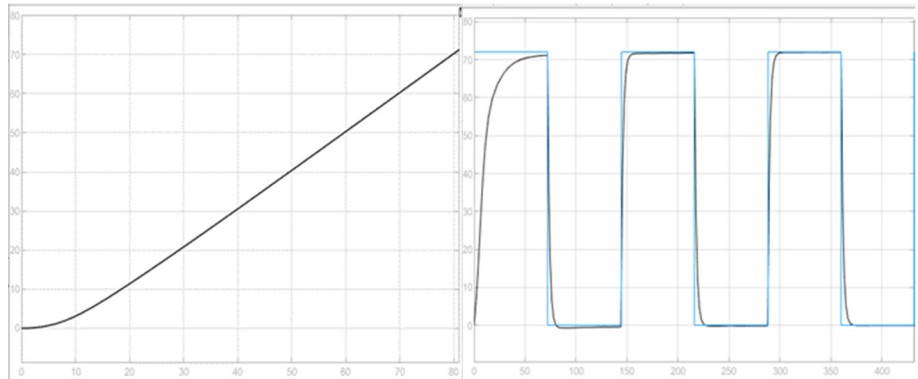

**Figure 38.** Response of the packaging process to a ramp input and a pulse generator, n = 7.

For the output behavior of a ramp type input you have an 80 min set time for the goal you want, and for a step type input you have on average one piece per minute, on average, for the steady state reach. With a generator of several input steps in several shekels, each cycle is faster and stable, because it is simulated in other processes with a single cycle time. With the ramp type inputs you get a more real behavior than what happens in a textile plant.

### 3.8. General Approach Proposed

3.8.1. Forrester Diagram-(n)-State Variables

The Forrester diagram is presented for n state variables, with 3n auxiliary variables, n flow variables, and n Stock. See Figure 39.

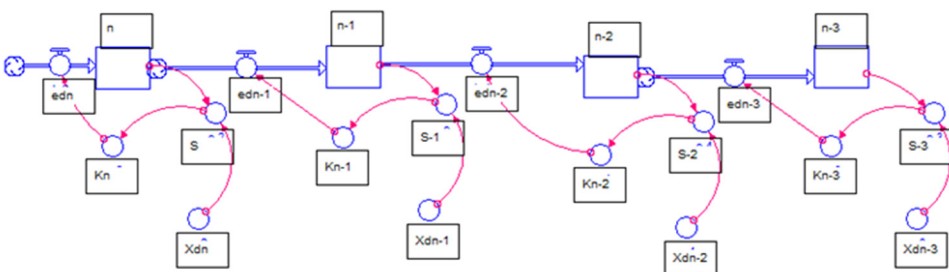

**Figure 39.** Forrester diagram of n, state variables.

It is identified that the state variable n, is subtracted from the flow variable $n_{-1}$ and it has the differential equation in time of the variable which must have been solved $n_{-1}(t)$. See Equation (18).

$$\frac{dn}{dt} = k_n(Xd_n - n) - k_{n-1}(Xd_{n-1} - n_{-1}(t))$$

3.8.2. Block-n-Process Diagram

For n state variables, it is of great importance that the resolution of the transformation of the Forrester diagrams to the frequency domain takes place, due to the fact that the differential equations become more difficult to solve and are quite late processes, even with specialized software. Therefore, for n differential equations or n state variables, it is advisable to carry out the research proposal; the diagram for n state variables is from left to right in the order of the processes, as in all the raised methodology. See Figure 40.

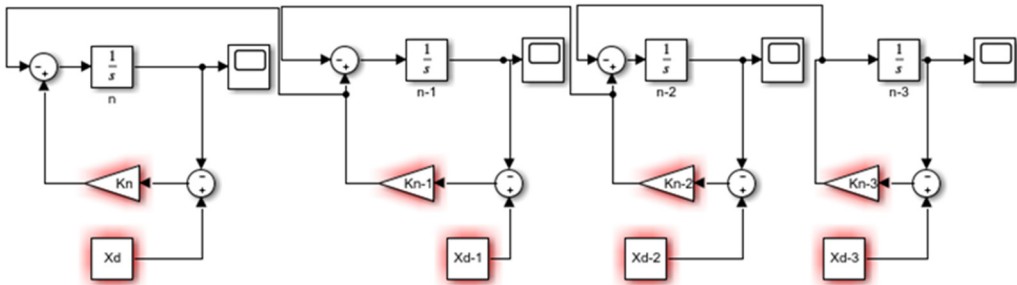

**Figure 40.** Block diagram of n, state variables.

3.8.3. Stability of the System-n-State Variables

From the transfer function of the system, the poles and zeros of the system can be identified; the poles of the system correspond to the eigenvalues of the system. We can define a system as stable when its output is bounded; that is, its output is not $\pm\infty$ but a concrete value. We could also define as stable that it evolves in a similar way to the input variable. The calculation of the stability of the system is equal to zero and the characteristic equations of the numbering and denominator (the eigenvalues) are taken out and plotted on a real and imaginary plane. Where the poles are in the left half-plane, it is a stable process, and if not, the process would be unstable.

3.8.4. Multiple Inputs-n State Variables

In general, when obtaining the transfer function for n state variables of n manufacturing processes, having the mathematical model with the transfer function is of great help for the behavior of multiple inputs to the system. Solving it by means of differential equations

manually, or even with software, would be quite complicated and tedious, when n can be a high number of state variables.

## 4. Results

The methodology proposed for the seven SD models was implemented and the following transfer functions were obtained for each of the state variables, as shown in Table 3.

**Table 3.** Transfer function of the textile process.

| State Variables. | Transfer Function |
|---|---|
| a | $H_a(s) = \frac{0.009541}{(s + 0.009541)}$ |
| b | $H_b(s) = \frac{0.1104s + 0.001145}{s^2 + 0.1295 + 0.01145}$ |
| c | $H_c(s) = \frac{0.001s^2 + 0.01338s + 0.0001385}{s^3 + 0.2505s^2 + 0.01682s + 0.0001385}$ |
| d | $H_d(s) = \frac{0.33s^3 + 0.08235s^2 + 0.007098s + 6.182\times10^{-5}}{s^4 + 0.6995s^3 + 0.129s^2 + 0.007648s + 6.183\times10^{-5}}$ |
| e | $H_e(s) = \frac{0.0001s^4 + 0.1488s^3 + 0.03728s^2 + 0.003221s + 2.805\times10^{-5}}{s^5 + 1.151s^4 + 0.4448s^3 + 0.06601s^2 + 0.003531s + 2.805\times10^{-5}}$ |
| f | $H_f(s) = \frac{0.05s^5 + 0.05748s^4 + 0.08901s^3 + 0.01996s^2 + 0.0016134s + 1.391\times10^{-5}}{s^6 + 1.65s^5 + 1.019s^4 + 0.2876s^3 + 0.03635s^2 + 0.001779s + 1.391\times10^{-5}}$ |
| g | $H_g(s) = \frac{0.05s^6 + 0.1075s^5 + 0.07967^4 + 0.05889s^3 + 0.0118s^2 + 0.0008953s + 7.651\times10^{-5}}{s^7 + 2.2s^6 + 1.926s^5 + 0.8478s^4 + 0.1945s^3 + 0.02177s^2 + 0.0009926s + 7.651\times10^{-5}}$ |

By obtaining the transfer function for each of the seven state variables, the output of the system can be obtained for a step-type input with amplitude of 72, which are equivalent to the goal of the Forrester diagram. In each of the outputs of the seven processes, the expected results are obtained, which are identical, with an error of less than 0.0001% with respect to the simulations in Stella. Besides, the behavior is expected in the time cycles of each of the processes of the textile sector. Another of the great advantages of the transfer function is that the type and value of input can be varied and the output can be obtained according to multiple inputs, either separately, or under the principle of superposition.

## 5. Discussion

The research methodology begins with the mapping of the n operations, which are involved in the manufacturing process, which for the case study is the textile process. There are seven state variables. Having the number of operations, or state variables, the Forrester model is proposed in Stella, posing each process as balance loops that seek a goal in the system. The processes are proposed from right to left. The differential equations that represent the manufacturing process are identified and from the differential equations the block diagrams in the Laplace domain are proposed. The block diagram is solved by proposing equations based on a signal treatment and the transfer function that relates to the input with the output of the system is found. With the output of the system, an analysis of the stability of the system is performed by means of the diagram of the poles and zeros of the system, and it is determined by the process with the eigenvalues that the process is stable.

The works presented by [23–34] perform an internal analysis of the dynamic model. Where they particularly perform an analysis of eigenvalues and eigenvectors as support, they explain the dominant structure of the system based on the dynamic behavior of the loops present in the SD model, but they do not perform an analysis of the stability of the system through the eigenvalues of the system. Additionally, the poles of the transfer function are the controllable poles of the dynamic process. Another advantage of the project performed is to find that transfer function that relates as an external analysis, due to multiple inputs, and to see how they affect the behavior of the system raised.

As a future project, it would be very interesting and of great help in the contribution of the mathematical formality of dynamic systems, to obtain the geometric location of the roots to control the limits of the values that can modify our system, from that of robust

analysis implemented in control engineering in multiple applications. Another future project would be to perform the analysis of system dynamics models by modern state space control theory.

## 6. Conclusions

The present research proposes an analytical alternative of analysis for a dynamic system applied in the textile sector, which can be replicated in various DS models. This is a mathematical model capable of explaining the control and behaviors of the state variables, which explains their external dynamics, and controls the trajectories of the simulation model, and is not only based on understanding the internal dynamics of the DS models. Another great advantage is to determine the transfer function of each of the state variables involved in the dynamic process; with this transfer function we can analyze the stability of the system.

The stability of the system was determined by means of the pole and zero diagram for each of the simulation models of the seven state variables finding stable processes, because the poles, or eigenvalues, of the system are in the left half-plane of the diagram. In addition to this, the transfer function can determine the output of the system to different inputs or external disturbances as to how the process would behave the manufacturing process. Another major challenge achieved is to demonstrate the transformation of Forrester models to block diagrams, commonly implemented in control engineering to make the comparison that the transformation is adequate for the output results observed with both diagrams. As future projects, the study of the geometric place of the roots can be carried out to determine the limits of the eigenvalues of the manufacturing system, or to replicate it in different simulation models of system dynamics; besides being able to perform a robust control by means of the modern control state space technique, to control the value of the eigenvalues and limits of stability or instability of the system.

It can be concluded that the type of input that reflects the reality of the textile and manufacturing process is the ramp type, since it is possible to observe the process in a linear way in how the objectives of different orders of a manufacturing company work.

**Author Contributions:** Conceptualization, R.B.S.; Methodology, J.M.B.S. and R.B.S.; Software, J.M.B.S. and R.B.S.; validation, J.M.B.S. and R.B.S.; formal analysis, J.M.B.S. and R.B.S.; investigation, J.M.B.S. and R.B.S.; resources, J.M.B.S., R.B.S. and M.B.; data curation, J.M.B.S. and R.B.S.; writing—original draft preparation, J.M.B.S. and R.B.S.; writing—review and editing, J.M.B.S., R.B.S. and M.B; All authors have read and agreed to the published version of the manuscript.

**Funding:** This research received no external funding.

**Informed Consent Statement:** Not applicable.

**Data Availability Statement:** The Stella and Simulink software simulation files were uploaded at the corresponding link.

**Conflicts of Interest:** The authors declare no conflict of interest.

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
