# Peer review of "Design and Development of a Mathematical Model for an Industrial Process, in a System Dynamics Environment"

_applsci, doi:10.3390/app12199855_

Round 1

Reviewer 1 Report

The article is very interesting in comparing results generated using different software and how this allows observing the modes of behavior in complex environments.

- Line 27, 139: Must start with a capital letter...the dS...The DS

- Some paragraphs are longer than 15 lines, it is recommended to separate them according to defined rules.  For example, the paragraph written from line 56 to line 80 can be separated to improve the communication of the idea.

- Explain the use of the phrase system dynamics and Dynamic systems in a similar way? Within system dynamics, the concept should be differentiated. According to the article it should be used as System Dynamics and corrected throughout the paper.

- Line 83: Correct Ds for DS, 

- Line 84: authors cited as [7-22] would indicate that they are 15 authors on the sentence: Today there are thousands of essential applications in the environmental, business, 82 social and scientific sector, developing Ds simulation models with a great impact at in-83 industrial and academic level, such as [7-22] developed models of DS presented in the field 84 of energy, medicine, agriculture, population development, industry 4.0 and sustainability, 85 of great value.

If you mean only 7 and 22 should be quoted [7,22].

- Line 155 defines CO please when first used.

- Line 161 should start with a capital letter the word Reason

- Line 23: table 2 should be quoted as Table 2

- Line 23: begin with a capital letter the word 2.1 Methodology

- Line 273, 433: correct the word forrester for Forrester and response for Response in the citation of Figure 13...Figure 20 should also be corrected Block. (check all cases where this spelling problem occurs)

- All paragraphs that start in Line 632-637 Numbers 1 to 10 are written with letters except in some situations, check this in your article please (check 3.5.1 should say five flow variables..... and not 5 flow variables).... 

- Same situation for paragraph lines 661-666; 687-690; 714-718.

Additional comments: 

I recommend reviewing the graphical and spelling aspects.

Explain the validation process of the model or models proposed: extremes or unit consistency test when using Stella for example, or the use of the Barlas test.

Author Response

Response to Reviewer 1 Comments

Good afternoon! It is reported that the document was adjusted according to each of the two reviewers' reservations and that some lines of reference in the suggestions have been changed.

Point 1: Line 27, 139: Must start with a capital letter...the dS...The DS

Response 1:

Following the suggestion to validate the acronym and give consistency throughout the document is reported that was adjusted the entire document leaving System Dynamics (SD)

SD is based on the use   (27)

            in the SD   (139)

Example of the adjustment document

  1. Introduction

System Dynamics (SD) is a technique for constructing simulation models, at the service of a fundamental idea: the structure of causal relations between the elements of a system constitutes the main cause of its behavior. The SD further assumes that such a relationship structure forms a network of re-feeding loops that includes non-linear relationships and delays. The main result of a model is to obtain the time trajectories of all its variables for the simulation period that has been defined [1]. SD is based on the use of two types of diagrams, Causal Diagrams and Forrester Diagrams, have their origin in the General Theory of Systems and are in fact like the two sides of the same coin. Causal Diagram is in general a preliminary step to the construction of a Forrester Diagram, which serves to simulate the model in the PC, allows us to verify the coherence of our hypotheses, analyze the behavior of the system and finally simulate different policies, so that the results of the model help to better solve the problem we are analyzing [2].

Point 2: Some paragraphs are longer than 15 lines, it is recommended to separate them according to defined rules.  For example, the paragraph written from line 56 to line 80 can be separated to improve the communication of the idea.

Response 2:

In response to his suggestion, it is reported that the document was adjusted in its entirety by checking the content of each paragraph to a maximum of 15 lines.

Example:

It is of great importance and relevance the design, development and simulation of SD models in various areas of knowledge, identifying and making explicit the various behaviors based on feedback loops and interactions. Making decisions based on simple explanations of the observed results, to improve them, as described by each of the previous authors. However, for a better understanding of the model and better decision-making, it is important to analyses the model of system dynamics with a theoretical and mathematical basis to represent, explain and control the behaviors of the studied variables.

This research proposes a methodology based on control engineering, transforming the simulation model of system dynamics into a mathematical model expressed as a system transfer function. It is important to note that currently the theory of SD does not make use of the necessary mathematical analysis that is used in the theory of control engineering to design control systems. Both theories analyze dynamic systems from their perspective, which are complementary to strengthen decision-making of SD simulation models as proposed in this research. In summary, SD analyzes dynamic systems based on a simulation model where multiple variables interact to analyze their behavior and control engineering analyzes dynamic systems obtaining mathematical models to design control systems.

Point 3 :Explain the use of the phrase system dynamics and Dynamic systems in a similar way? Within system dynamics, the concept should be differentiated. According to the article it should be used as System Dynamics and corrected throughout the paper.

Response:

In response to this question, it is reported that different techniques exist to analyze dynamic systems. One is system dynamics and the other is control engineering. A couple of paragraphs were added to make the respective description as follows:

System Dynamics (SD) is a theory that analyzes dynamic systems (DS) based on simulation models. Control engineering analyzes dynamic systems (DS) based on mathematical models.

It is important to note that currently the theory of SD does not make use of the necessary mathematical analysis that is used in the theory of control engineering to design control systems. Both theories analyze dynamic systems from their perspective, which are complementary to strengthen decision-making of SD simulation models as proposed in this research. In summary, SD analyzes dynamic systems based on a simulation model where multiple variables interact to analyze their behavior and control engineering analyzes dynamic systems obtaining mathematical models to design control systems.

- Point 4. Correct Ds for DS

Response:

Again it is reiterated that adjustments were made throughout the document to the System Dyamics acronym such as SD.

Example

Today there are thousands of important applications in the environmental, business, social and scientific sector, developing SD simulation models with a great impact at industrial and academic level, such as [7-22] developed models of SD presented in the field of energy, medicine, agriculture, population development, industry 4.0 and sustainability, of great value. In the validation of all the models cited, they use the technique of sensitivity analysis to validate the models, generating multiple scenes and changes in critical variables subjectively and analyzing changes in their behavior. Since there are multiple variables and interactions in the systems dynamics models, and only the interpretation of the results obtained, it is difficult and complex to identify which parameters should be chosen to perform the sensitivity analysis.

It is evident an area of opportunity to define a mathematical model to identify parameters that affect behavior in a simple, analytical and objective way rather than the current subjective choice. The present research proposes an analytical alternative of analysis for a SD models, able to explain the behaviors of state variables, obtaining a mathematical model that explains its external dynamics and controls the particular trajectory of each behavior, instead of just observing the response without understanding whether or not there is any relation to the output in relation to the input of the model under study. In addition, by determining the transfer function, it is possible to analyse and monitor the response to multiple changes in the inputs to the system, the fundamental reason and motivation of this research project.

Point 5. Line 84: authors cited as [7-22] would indicate that they are 15 authors on the sentence: Today there are thousands of essential applications in the environmental, business, 82 social and scientific sector, developing Ds simulation models with a great impact at in-83 industrial and academic level, such as [7-22] developed models of DS presented in the field 84 of energy, medicine, agriculture, population development, industry 4.0 and sustainability, 85 of great value.

Response:

The interpretation of the citation is as follows: There are 15 authors with different investigations that refer to applications of system dynamics in various industrial sectors.

Waiting for your valuable feedback in case change is required.

Point 6.

Line 155 defines CO please when first used.

Response:

The corresponding adjustment was made by describing the full name before the acronym.

It should be noted that the authors only consider the first term of equation 1 in the reduction of the system dynamics model to a linearized system invariant in time, making incomplete analysis of the linearized model and reducing the analysis only to the internal dynamics of the  system. In addition, the technique used to analyze the internal dynamics of the system based on the matrix of partial derivatives in control engineering theory, has the objective of defining the internal stability of the system through the decomposition and interpretation of its characteristic canonical modes, which may belong to one of the following combinations: controllable and observable CO, not controllable and observable , controllable and not observable , not controllable and not observable , which are not mentioned, or discussed in the above researches and presents an area of opportunity to perform a complete analysis of the systems dynamics models and to strengthen the theory that supports the explanation of the system structure link-dynamic behaviour for decision making.

- Point 7. Line 161 should start with a capital letter the word Reason

Response:

The adjustment for the initial letter was made.

Reason for this research by proposing a methodology based on conventional control engineering theory, to determine a transfer function of the SD model based on the Laplace transform that explains the setting state variable trajectories to impulse or system inputs, transformed the time domain differential equations for solution to a frequency domain and analyze the external stability and controllability of the system based on the decomposition of controllable characteristic modes (poles) and observable (zeros) in the polynomial of the transfer function, to control and manipulate the behaviors of the model’s significant variables and more robust decision-making, in contrast to the research cited where only the dominance of feedback loops is analyzed and the analysis of the internal and external stability of the system based on the decomposition and interpretation of characteristic modes (eigenvalues) is not contemplated or the determination of a function that explains the particular trajectories to predict the response of the system to changes in the input variables, which as mentioned above is not concluded by not interpreting its result to determine whether the system is stable or not stable and not to generate the particular trajectory of each state variable as a system response.

Point 8. Line 23: table 2 should be quoted as Table 2

Response:

The adjustment for the initial letter was made.

The notations of the state variables, and representation of the differential equations can be represented in Table 2, which are useful and necessary for Forrester diagrams and Block diagrams. From a sensitivity analysis, the k parameters of each differential equation were determined. The values of k are based on the accumulated cycle times considering the number of machines in each of them. Each process is simulated in a general way with an added cycle time in order not to perform the simulation on each of the machines of each operation.

The level of production in each operation gives rise to a first order differential equation. If n(t) denotes the number of sweaters produced in the state variable at time t, then the speed at which n(t) changes is a net speed:

dn/dt=(input rate)-(output rate)

Table 2. Data on auxiliary variables and notation of state variables.

Point 9: Line 23: begin with a capital letter the word 2.1 Methodology

Response:

The adjustment for the initial letter was made.

2.1.  Methodology.

The methodology was carried out in 8 steps, which can be seen below, for the realization of this research project. See Figure 1.

- Point 10: Line 273, 433: correct the word forrester for Forrester and response for Response in the citation of Figure 13...Figure 20 should also be corrected Block. (check all cases where this spelling problem occurs)

Response:

The adjustment for the initial letter was made throughout the document.

Example:

Manufacturing process mapping: A mapping is made of the operations involved in the manufacturing process in studied and carried out ascendingly until reaching the n operations.

Making the Forrester diagram: The Forrester diagram is made in the Stella software, where the state or stock variables, flow variables, feedbacks and auxiliary variables involved in the manufacturing process are identified.

Differential equation identificaction: The differential equations of the simulation model are identified,  from the Forrester model, the differential equations are those that model mathematically and serve as support for the simulation model.

Transformation to Block diagram (S): The transformation of the Forrester diagram to the Block diagram is carried out by means of the differential equations that model the simulation system.

 For the reduction of the block diagram, the matlab simulink tool is used, where the design of the Block diagram is performed and the parameters of k and Xd1 are introduced, thus obtaining the transfer function of the system in equation 4.

- Point 10: All paragraphs that start in Line 632-637 Numbers 1 to 10 are written with letters except in some situations, check this in your article please (check 3.5.1 should say five flow variables..... and not 5 flow variables).

Response:

The adjustment of numbers expressed in words was made throughout the document.

There are fifteen auxiliary variables, five flow variables and five stocks. The complexity of obtaining a 5th state variable is not even manageable to be done by means of specialized software due to the number of differential equations to be performed, in the MATLAB software, to determine the function in time of the differential equation for the case study, it did not solve the solution in a suitable way as a function of time. On the other hand, there are five balancing loops that for the 5th state variable is affected by the whole system there is a fast decrease of the process in the non-operation time of the process. See Figure 24.

Same situation for paragraph lines 661-666; 687-690; 714-718.

The polynomial equals to zero and the roots of the numerator and denominator are taken out, there are three zeros and five poles, this is because the fourth and fifth state variable have the same k, that is to say at the beginning of the process of the 5th state variable, it does not start the accumulation because initially what enters and leaves the flow variables ed5 and ed4 is the same.

Resulting five poles on the real axis, two imaginary conjugate zeros and a zero on the real axis, all in the left plane of the pole diagram and resulting in a stable system for the five state variables, where the poles of the system are equal to the value of the k, assigned in the Forrester diagram. See Figure 27.

There are eighteen auxiliary variables, six flow variables and six stocks. The complexity of obtaining a 6th state variable is not even manageable to be done by means of specialized software due to the number of differential equations to be performed, in the MATLAB software, to determine the function in time of the differential equation for the case study, it did not solve the solution in a suitable way as a function of time. On the other hand, there are six balancing loops that for the 6th state variable is affected by the whole system there is a fast decrease of the process in the non-operation time of the process. See Figure 29.

Additional comments:

Point 11: I recommend reviewing the graphical and spelling aspects.

Response:

The orthographic revision adjustment was made for the entire document as well as the figures.

Point 12: Explain the validation process of the model or models proposed: extremes or unit consistency test when using Stella for example, or the use of the Barlas test.

Response:

The document cites how to perform validation through sensitivity analysis.

Having the construction and understanding of the SD model, currently the validation of the models is performed by means of a sensitivity analysis, which consists of a systematic study of how the conclusions of a model are affected by possible variations in parameter values and the functional relationships it includes. The simplest way to perform the analysis is to modify the numerical values of each of its parameters. For this purpose, the value of the parameter whose sensitivity is to be studied is increased by a certain percentage and the extent to which this variation affects the conclusions of the model (the trajectories it generates) is analyzed. Manual sensitivity analysis requires changing the value of a constant (or several constants at once) and simulating, changing the value of the constant again and simulating again and repeating this action many times to achieve a spectrum of output values [6].

In the research proposal, the validation of the model is performed in numerical form through the mathematical model of the transfer function that allows us to obtain the trajectory of each of the state variables, as is the main objective of the research.

It is evident an area of opportunity to define a mathematical model to identify parameters that affect behavior in a simple, analytical and objective way rather than the current subjective choice. The present research proposes an analytical alternative of analysis for a SD models, able to explain the behaviors of state variables, obtaining a mathematical model that explains its external dynamics and controls the particular trajectory of each behavior, instead of just observing the response without understanding whether or not there is any relation to the output in relation to the input of the model under study. In addition, by determining the transfer function, it is possible to analyse and monitor the response to multiple changes in the inputs to the system, the fundamental reason and motivation of this research project.

Reviewer 2 Report

Will be appropriate to clearly formulate the novelty of the proposed approach, because application of transfer functions is good known approach to research system dynamics.

The order of the numerator of the transfer function is n-1, and the denominator is n. And only for systems of the 3rd and 5th orders, this regularity is not fulfilled (see Eq. (8) and (12)). Probably, that arose as a result of neglected the very small coefficient with the term in power n-1. It is appropriate to justify such reduction of the numerator polynomial order.

Diagram 1 describes the always stable first order system. Therefore, the study of the stability of the system conducted by the authors is unclear. 

What is the expediency of investigating the stability of the system based on the poles of the transfer function if  the  transient characteristic is known?

Would be appreciable to clarify different dynamics for step response on fig. 13 or fig. 38, for example

It would be appropriate to explain, how the parameters of separate process models are determined

Author Response

Response to Reviewer 2 Comments

Point 1: Will be appropriate to clearly formulate the novelty of the proposed approach, because application of transfer functions is good known approach to research system dynamics.

Response:

In response to this question, it is reported that different techniques exist to analyze dynamic systems. One is system dynamics and the other is control engineering. A couple of paragraphs were added to make the respective description as follows:

System Dynamics (SD) is a theory that analyzes dynamic systems (DS) based on simulation models. Control engineering analyzes dynamic systems (DS) based on mathematical models.

It is important to note that currently the theory of SD does not make use of the necessary mathematical analysis that is used in the theory of control engineering to design control systems. Both theories analyze dynamic systems from their perspective, which are complementary to strengthen decision-making of SD simulation models as proposed in this research. In summary, SD analyzes dynamic systems based on a simulation model where multiple variables interact to analyze their behavior and control engineering analyzes dynamic systems obtaining mathematical models to design control systems.

Point 2: The order of the numerator of the transfer function is n-1, and the denominator is n. And only for systems of the 3rd and 5th orders, this regularity is not fulfilled (see Eq. (8) and (12)). Probably, that arose as a result of neglected the very small coefficient with the term in power n-1. It is appropriate to justify such reduction of the numerator polynomial order.

Response 2:

It is correct what you point out in your observation and that happens because the values of k in two state variables are identical and is reflected in the cancellation of a zero in the numerator.

To solve their observation, slight increments were made in the constants k to obtain the generalization of the order n in the denominator and n-1 in the numerator.

See Transfer functions in Table 3.

Table 3. Transfer function of the textile process.

State Variables.

          a

b

c

d

e

f

g

                                   H(s) 

Point 3: Diagram 1 describes the always stable first order system. Therefore, the study of the stability of the system conducted by the authors is unclear.

Response :

In response to the comment, a paragraph was inserted to explain more clearly.

3.1.4 Stability of the knitting process system.

The behavior of Figure 6 is of a first order difference equation. It is important to note that the time response of a control system consists of two fundamental parts: the transient response and the steady state response. Every first-order system has 1 pole which governs its dynamics. If that pole is close to the imaginary axis, it makes the response of the system much slower (that is, its transitory state will take longer) If the pole is far from the imaginary axis, the response of the system will be fast ( fast transient state). The transfer function obtained shows us a behavior without delays.The transient response is affected by the pole of the transfer function, but the moment it reaches steady state does not depend on the pole of the system. The pole for the transfer function obtained is 0.009541 which is very close to zero, which represents a slow transient to reach the stable state.

Point 4: What is the expediency of investigating the stability of the system based on the poles of the transfer function if  the  transient characteristic is known?

Response:

In response to the observation, a paragraph was added explaining the origin of the state variable transient’s behavior. It should be noted that currently there is no research in simulation theory that validates through a mathematical model.

It should be noted that the transient characteristic is known from Forrester simulation models, based on the system dynamics methodology. This transient characteristic is calculated by numerical methods by simulation in the Stella program, for example the euler and newton raphson method.

Later, with the proposed research, the Forrester diagram is transformed into Block diagrams to obtain the respective transfer function and it is checked by the poles and zeros obtained by the transfer function that the system is stable.

Point 5: Would be appreciable to clarify different dynamics for step response on fig. 13 or fig. 38, for example.

Response:

The graphs were better explained as suggested in the observation.

The clearance or follow-up time to reach the goal with a ramp entry is about 10% longer, which is logical because the process is followed and there are delays between real and recorded. The step input in its first cycle has a faster behavior than the previous state variable, but when modeling two or more cycles the transient response is even faster, because we take into account that it is not less affected by the previous state variable. See Figure 13.

Figure 18: The preparation time is 176 minutes about 15% longer than the time needed to reach the target, the delay increases due to the delay of the previous departments by about 5% with the immediately previous one. The transient response is slower than in the previous process, you have an establishment time in 144 minutes approximately.

Figure 23: The cutting process has a faster response when reaching the 72 pieces as the objective of the manufacturing process, with different inputs the process will be faster with respect to the previous processes, for multiple inputs, a setup time of 57 minutes is obtained, that is to say, besides being the process with the fastest response of all before a step input, it also takes approximately 5% to follow up the process. There is a quick exit response for a ramp entry with a linear slope of 1,333 pieces per mi-nute, the response is fifty-seven minutes and the step entry is the fastest transient of all transfer functions presented, in about fifty-four minutes.

 It has a set time of 84 minutes approximately for the first half of the pulse train period and descends in the dead period, the ramp type setting time is 10% slower than the step. See Figure 28.

For a ramp entry you get a response time of approximately eighty minutes for the record-ing of the seventy-two pieces, while for a step type input you have a transient of seven-ty-two minutes and a speed increase in the second cycles of the process, because in the seconds or n cycles you take only one cycle of the previous processes. See Figure 33.

.

Figure 38: For the output behavior for a ramp type input you have an eighty minute set time for the goal you want and for a step type input you have on average one piece per minute on average for the steady state reach. With a generator of several inputs step in several shekels has that in each cycle is faster stable, because it is simulated that other processes in a single cycle time. With the ramp type inputs you get a more real behavior than what happens in a textile plant.

Point 6: It would be appropriate to explain, how the parameters of separate process models are determined

Response:

To explain the determination of the main parameters a couple of paragraphs were added describing the meaning and the tables were adjusted for better understanding.

Materials and Methods

The case study to implement the proposed methodology is the knitting textile process, which consists of 7 process operations. The main parameters to model each of them is their respective cycle times, that is, the time spent in making a sweater. See Table 1.  Each of the different departments represent a state variable. Establishment time is determined for a production of 72 sweaters considering the number of machines available in each process operation.

Table 1. Cycle times of the textile manufacturing process

State variable

Cycle time (minutes)

Number of machines

Establishment time for 72 sweaters

knitting

40

6

480

Basting

3

2

108

Ironing

4

2

144

Cutting

3

4

54

Making

7

6

84

Finishing

2

2

72

Packing

2

2

72

The notations of the state variables, and representation of the differential equations can be represented in Table 2, which are useful and necessary for Forrester diagrams and Block diagrams. From a sensitivity analysis, the k parameters of each differential equation were determined. The values of k are based on the accumulated cycle times considering the number of machines in each of them. Each process is simulated in a general way with an added cycle time in order not to perform the simulation on each of the machines of each operation.

The level of production in each operation gives rise to a first order differential equation. If n(t) denotes the number of sweaters produced in the state variable at time t, then the speed at which n(t) changes is a net speed:

dn/dt=(input rate)-(output rate)

Table 2. Data on auxiliary variables and notation of state variables.

State variable (m)               k

Notation of 

the Variable

Differential equations

Knitting

0.009541

a

da/dt=k1(Xd1-a)

Basting

0.12

b

db/dt=k2(Xd2-b)-k1(Xd1-a)

Ironing

0.12

c

dc/dt=k3(Xd3-c)-k2(Xd2-b)

Cutting

0.45

d

dd/dt=k4(Xd4-d)-k3(Xd3-c)

Making

0.45

e

de/dt=k5(Xd5-e)-k4(Xd4-d)

Finishing

0.5

f

df/dt=k6(Xd6-f)-k5(Xd5-e)

Packing

0.55

g

dg/dt=k7(Xd7-g)-k6(Xd6-f)

Round 2

Reviewer 2 Report

Dear authors! Thank You for your response.

1. From the point of view of classical Control Theory, this is a fairly simple task that does not require detailed explanations.

For example, for the first order system:

- system is stable when all coefficient is positive;

- transfer function 0.009541/(s+0.009541) can be rewritten as 1/(T*s+1), where T=1/0.009541,  and steady state obtain at   3 … 5  time constant T;

- the value in the steady state can be estimated by substituting s=0 into the transfer function.

 2. According to You response: "The step input in its first cycle has a faster behavior than the previous state variable, but when modeling two or more cycles the transient response is even faster, because we take into account that it is not less affected by the previous state variable. See Figure 13", I must indicate, that transient dynamics depend from zero conditions and parameters of transfer function. In order to obtain such transient process, it is necessary to change the transfer function at the beginning of the second cycle in order to cancel the influence of the previous process. And it would be appropriate to indicate this transfer function.

3. There are some mistakes at line 694, 764, 827. The n-order system cannot has the n+1 poles.

Author Response

Response to Reviewer 2 Comments

Dear Reviewer, thank you very much for your revisions, this document makes the adjustments requested

Point 1:  From the point of view of classical Control Theory, this is a fairly simple task that does not require detailed explanations.

For example, for the first order system:

- system is stable when all coefficient is positive;

- transfer function 0.009541/(s+0.009541) can be rewritten as 1/(T*s+1), where T=1/0.009541,  and steady state obtain at   3 … 5  time constant T;

- the value in the steady state can be estimated by substituting s=0 into the transfer function.

Response:

It is reported that the suggestion made was modified in a clear and simple way as the reference, being as follows:

For the first order system represented by the transfer function obtained, a positive coefficient is obtained which indicates that the system is stable and by equalizing the denominator of the function, a real pole located in the negative left semiplane is obtained. See Figure 7.

Figure 7. Diagram of poles and zeros weaving process.

The establishment time to reach the target by 98% consists of 4 times the value of T. For the system the establishment time is 419.28 minutes. To get the value in the stationary state you do s=0 and you get that it is one for the first order mathematical function.

  1. Point 2: According to You response: "The step input in its first cycle has a faster behavior than the previous state variable, but when modeling two or more cycles the transient response is even faster, because we take into account that it is not less affected by the previous state variable. See Figure 13", I must indicate, that transient dynamics depend from zero conditions and parameters of transfer function. In order to obtain such transient process, it is necessary to change the transfer function at the beginning of the second cycle in order to cancel the influence of the previous process. And it would be appropriate to indicate this transfer function.

A couple of paragraphs were added to explain the suggestion and the transfer function that generates the behaviors of cycles greater than 1 was added.

For the second and third cycle the establishment time is faster than the first cycle. The process has initial conditions of 0 and does not have the influence of the knitting process. The transfer function that governs the behavior of any cycle greater than 1 is in equation (7).

Where the constant time T of the system transfer function is 8.333. The establishment time to reach 98% of the target is 33.33 minutes.

  1. Point 3: There are some mistakes at line 694, 764, 827. The n-order system cannot has the n+1 poles.

The respective adjustment is made in the document on the suggested lines, where a transfer error was made when repeating the -0.12 pole three times and the correct are twice, When making the adjustment the number of poles already coincides with the number of state variables.

                                                                     Line 694

poles = -0.451, -0.45, -0.12, -0.12 and -0.0095

               Line 764

poles = -0.5, -0.45, -0.45, -0.12, -0.12 and -0.0095

                Line 827

  poles = -0.55, - 0.50, -0.45, -0.45, -0.12, -0.12 and -0.0095
